



# Late-Spring and Summertime Tropospheric Ozone and NO2 in Western Siberia and the Russian Arctic: Regional Model Evaluation and Sensitivities.

Thomas Thorp[1], Stephen R. Arnold[1], Richard J. Pope[1,2], Dominic V. Spracklen[1], Luke Conibear[1], Christoph Knote[3], Mikhail Arshinov[4], Boris Belan[4], Eija Asmi[5], Tuomas Laurila[5], Andrei Skorokhod[6], Tuomo Nieminen[7], Tuukka Petäjä[7].

[1] Institute for Climate and Atmospheric Science, School of Earth and Environment, University of Leeds, Leeds, LS2 9JT, UK.
[2] National Centre for Earth Observation, University of Leeds, Leeds, LS2 9JT, UK.
[3] Meteorological Institute, Ludwig- Maximilians-University Munich, Theresienstr. 37, 80333 Munich, Germany.
[4] V.E Zuev Institute of Atmospheric Optics, Russian Academy of Sciences, Siberian Branch, Tomsk, Russia.
[5] Finnish Meteorological Institute, Climate Research Programme, 00101, Helsinki, Finland.
[6] A.M Obukhov Institute of Atmospheric Physics, Russian Academy of Sciences, Moscow, Russia.
[7] Institute for Atmospheric and Earth System Research, University of Helsinki, Finland.

## Abstract

We use a regional chemistry transport model (WRF-Chem) in conjunction with surface observations of tropospheric ozone and Ozone Monitoring Instrument (OMI) satellite retrievals of tropospheric column NO2 to evaluate processes controlling the regional distribution of tropospheric ozone over Western Siberia for late-spring and summer in 2011. This region hosts a range of anthropogenic and natural ozone precursor sources, and serves as a gateway for near-surface transport of Eurasian pollution to the Arctic. However, there is a severe lack of in-situ observations to constrain tropospheric ozone sources and sinks in the region. We show widespread negative bias in WRF-Chem tropospheric column NO2 when compared to OMI satellite observations from May – August, which is reduced when using ECLIPSE v5a emissions (FMB= -0.82 to -0.73) compared with the EDGAR-HTAP-2 emissions data (FMB= -0.80 to -0.70). Despite the large negative bias, the spatial correlations between model and observed NO2 columns suggest that the spatial pattern of NOx sources in the region is well represented. Based on ECLIPSE v5a emissions, we assess the influence of the two dominant anthropogenic emission sectors (transport and energy) and vegetation fires on surface NOx and ozone over Siberia and the Russian Arctic. Our results suggest regional ozone is more sensitive to anthropogenic emissions, particularly from the transport sector, and the contribution from fire emissions maximises in June and is largely confined to latitudes south of 60N. Large contributions to surface ozone from energy emissions are simulated in April north of 60N, due to emissions associated with oil and gas extraction. Ozone dry deposition fluxes from the model simulations show that the dominant ozone dry deposition sink in the region is to forest, averaging 6.0 Tg of ozone per month, peaking at 9.1 Tg of ozone deposition during June. The impact of fires on ozone dry deposition within the domain is small compared to anthropogenic emissions, and is negligible north of 60°N. Overall, our results suggest that



surface ozone in the region is controlled by an interplay between seasonality in atmospheric transport patterns, vegetation dry deposition, and a dominance of transport and energy sector emissions.

## 1. Introduction

In recent decades, the high latitudes have warmed disproportionately relative to global mean temperature increases, resulting in rapid environmental changes in the Arctic region, most notably substantial loss of summer sea ice (IPCC, 2014). This disproportionate warming is termed Arctic Amplification, and results from efficient Arctic feedback processes, such as surface albedo and temperature feedbacks (Pithan and Mauritsen, 2014). Although Arctic warming has been predominantly controlled by radiative forcing from well mixed greenhouse gases, such as carbon dioxide ($CO_2$), warming from changes in the abundances and distributions of short-lived climate pollutants (SLCPs) such as tropospheric ozone and aerosol particles may have contributed substantially (Sand et al., 2016). Targeting such SLCPs, through short-term emission controls could have a substantial benefit in mitigating Arctic and global warming, particularly in the near-term (Shindell et al., 2012).

Tropospheric ozone ($O_3$) is a secondary pollutant, and an SLCP, being a greenhouse gas with an atmospheric lifetime of several weeks (Stevenson et al., 2006). Tropospheric ozone is formed through photochemical oxidation of volatile organic compounds (VOCs), in the presence of nitrogen oxides ($NO_x$ = NO + $NO_2$) and sunlight (Crutzen et al., 1999). Enhancements in near-surface ozone degrade air quality, and are linked with premature mortality in humans (Atkinson et al., 2016; Jerrett et al., 2009; Lelieveld et al., 2015; Turner et al., 2016). Ozone is also detrimental to natural vegetation and crops (Fuhrer, 2009; Hollaway et al., 2012; Rydsaa et al., 2016), and can indirectly impact climate and hydrology through its impacts on vegetation carbon sequestration (Sitch et al., 2007) and transpiration (Arnold et al., 2018). Sources of tropospheric ozone and its precursors are poorly characterised in the Arctic region, resulting in poor understanding of sensitivity of ozone and its impacts to potential changes in Arctic atmospheric processes, and remote and local emission sources (Law et al., 2017). Local Arctic sources of ozone precursors may increase in the future with northward migration of population, an expanding tourism industry, and increased industrial activity and shipping traffic (Arnold et al., 2016; Schmale et al., 2018).





Western Siberia is an important region in the context of high latitude tropospheric ozone concentrations, as it possesses an array of potential precursor sources. During winter and spring, the region acts as a "gateway" for poleward near-surface advection of Eurasian pollution into the Arctic (Stohl, 2007), which contributes to the well-characterised "Arctic haze" (Shaw, 1995; Quinn et al., 2008). However, a severe paucity of in-situ observations limits our understanding of sources, sinks and processing of pollution over Western Siberia, including ozone and its precursors. Current emission inventories have large uncertainties for high latitude emissions, including those from Western Siberia (Schmale et al., 2018).

The Western Siberia region is impacted by both anthropogenic and natural ozone precursor sources, many of which are poorly quantified. Anthropogenic sources in the region include those associated with large urban regions such as transport, domestic heating and power generation (Stohl et al., 2013), as well as sources specific to industrial and commercial activities in the region, such as gas flaring (Huang et al., 2014; Huang et al., 2015; Marelle et al., 2018) and shipping (Corbett et al., 2010). Moreover, future emission increases are likely, meaning a better understanding of these sources is important in the context of future Arctic SLCP budgets (Arnold et al., 2016). The Ob Valley region in particular is home to multiple populous cities, such as Novosibirsk (1.5 million people), Yekaterinburg (1.4 million), Novokuznetsk (550,000) and Tomsk (550,000). Emissions from these urban regions are uncertain, and poorly constrained by in-situ monitoring, except for some long-term datasets reported for Tomsk by Davydov et al., (2019). Vivchar et al., (2009) used a back-trajectory model to quantify local source regions of $NO_x$ emissions from Siberia to the Zotino observation tower (Zotino location shown in Fig. 1). Their results suggest a significant contribution to $NO_x$ pollution levels found in the background Siberian atmosphere originating from sources to the south of Zotino, which includes regions of intense pollution, such as the Ob Valley area.

Past studies have attempted to improve quantification of several ozone precursor emission sources in Western Siberia. Low light imaging data from the Defence Meteorological Satellite Program (DMSP) suggest that the volume of gas flared decreased between 2005 and 2008 (Elvidge et al., 2009), however a





study examining tropospheric column NO2 specifically from gas flaring locations in Western Siberia found no significant trends between 2004-2015 (Li et al., 2016). Improvements in flaring efficiency is most likely the reason for the observed decrease. High latitude residential combustion (Stohl et al., 2013) and transport emissions (Huang et al., 2015) are often underestimated, or overlooked entirely in this

region. Residential combustion at the latitudes relevant to the Ob Valley (55°N - 65°N), can result in emissions all year round which relates to indoor heating and cooking, due to prolonged wintertime low outdoor temperatures, and frequent summer cold spells. The use of diesel generators to provide the energy for this heating is frequently understated, which may be used both domestically and commercially as space heaters for up to 12 hours a day (Evans et al., 2015). Attempts to better quantify Russian transport

sector emissions suggest major flaws in current emissions. In particular a severe lack of regional activity data, a problem shared across all major anthropogenic sectors, leads to missing contributions from major sources. This is highlighted by Evans et al., (2015), where a detailed inventory is provided for Murmansk in northern European Russia, which is the largest city within the Arctic Circle. Murmansk is recognised as a particular region of poor emission quantification due to high levels of industrial mining, which is

often overlooked (Stohl et al., 2013). A coherent evaluation of anthropogenic ozone precursor sources across the region is lacking.

In addition to fossil combustion sources, during summertime, large wild and agricultural fires emit substantial amounts of ozone precursor species (AMAP, 2015), and are the largest natural source of

pollutants from within the Arctic region (Schmale et al., 2018). The intensity and location of these fires vary annually, but the frequency of high impact Siberian fire events is increasing (Kukavskaya et al., 2016). Over the past 20 years, severe Siberian fire-events occurred in 2003 (Jeong et al., 2008), 2010 (Konovalov et al., 2011), 2012 (Antokhin et al., et) 2014 (Jung et al., 2016) and 2016 (Sitnov et al., 2017). These fires can have severe impacts on regional air quality, and lead to increases in aerosol and ozone

concentrations, which can perturb the radiative budget, affecting regional climate. During a severe heat wave in 2010, which led to severe wildfires to the east of Moscow, 11,000 nonaccidental deaths were associated with increased levels of pollutants and degradations in air quality attributed to wildfires (Shaposhnikov et al., 2014). Understanding the controls on tropospheric ozone concentrations in a region




of wildfires can be further complicated, due to high levels of aerosols associated with fires (Jaffe and Wigder, 2012). In Siberia, this has been found to limit photochemical ozone production, and also act as an ozone sink in some cases (Antokhin et al., 2018).

5  Siberia is also characterised by extensive vegetation cover, which may act as an important dry deposition sink for pollution in the region. Previous studies have demonstrated observations of suppressed high latitude ozone concentrations in air masses that have had extensive surface contact with Siberian forests (Hirdman et al., 2010; Engvall et al., 2012). This implies a key role for Siberian vegetation as a sink for ozone pollution in the region, potentially reducing the abundance of ozone within air masses transported

10  polewards from ozone precursor source regions. An understanding of the extent to which this ozone sink mediates anthropogenic ozone influence in high latitude Siberia requires detailed quantification.

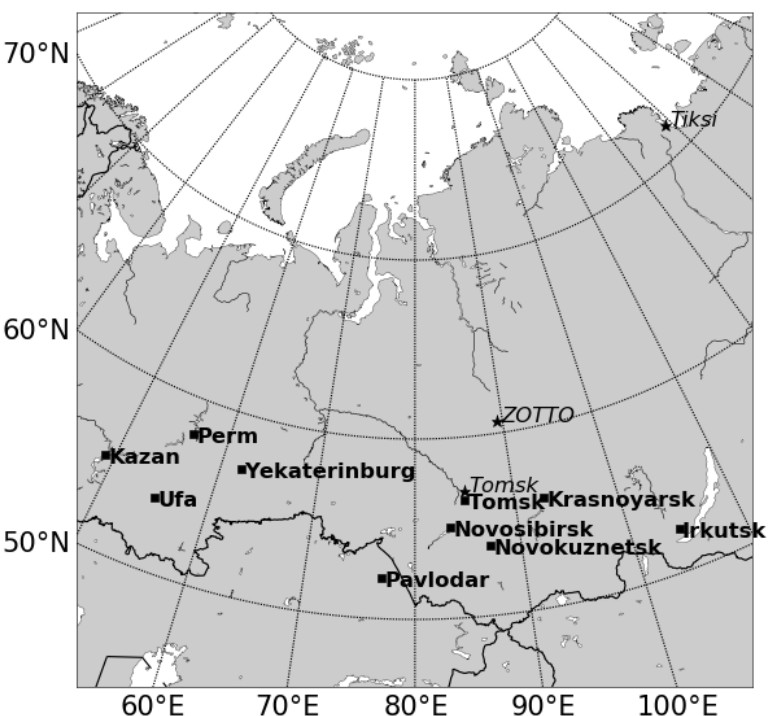

**Figure 1 -** Map of domain used for model simulations. Centred on Western Siberia region, major cities (squares) (population > 100,000) shown in bold. Observation sites (star symbols) are given in italics.

In this study we use satellite observations, surface measurements, and a regional air quality model to evaluate spring and summer tropospheric $NO_2$ and ozone in Western Siberia. Our model domain encompasses both the major Western Siberian cities to the south and the Arctic Ocean coast to the north, whilst also enabling us to capture potential springtime poleward transport of pollutants (Fig. 1). Our overall aim is to exploit satellite $NO_2$ observations to better understand sources of ozone precursors in a region of sparse in-situ measurements, and to investigate major processes controlling surface ozone in this region. We evaluate the performance of

two different commonly used anthropogenic emissions inventories in the region and use the model to quantify contributions to surface ozone from anthropogenic and vegetation fire precursor emissions. Finally, we use the model to estimate the contributions from different types and regions of vegetation in Western Siberia to dry deposition loss of ozone produced from anthropogenic and fire emissions from the region and ozone originating upstream. Section 2 introduces the methodology used within the study, section 3 presents the results, section 4 provides a discussion and section 5 finishes with the main conclusions from the study.

## 2. Data & Methodology

### 2.1 Anthropogenic Emission Inventories

We use and compare two different anthropogenic emission inventories: the EDGAR (Emissions Database for Global Atmospheric Research)-HTAP (Hemispheric Transport of Air Pollution) v2.2 inventory, and the ECLIPSE (Evaluating the Climate and Air Quality Impacts of Short-Lived Pollutants) V5a inventory. We carry out two model simulations to compare the impacts of these different emission datasets on ozone and its precursors in the Western Siberia region.

#### 2.1.1 EDGAR-HTAP V2.2

The EDGAR-HTAP v2.2 (hereafter "EH2") (Janssens-Maenhout et al., 2015) anthropogenic emissions used are for the year 2010 and acquired in a monthly 0.1° x 0.1° gridmap format and split into anthropogenic sectors (aircraft, shipping, energy, industry, transportation, residential and agriculture). 2010 EH2 emission species include: carbon monoxide (CO), sulphur dioxide ($SO_2$), $NO_x$, NMVOC, ammonia ($NH_3$), particulate matter smaller than 10µm ($PM_{10}$), particulate matter smaller than 2.5 µm ($PM_{2.5}$), black carbon (BC), organic carbon (OC) and methane ($CH_4$). The EH2 emissions are created through supplementing globally reported emissions with regional inventories, with the aim of producing





an inventory for hemispheric transport of air pollution. These data are readily available online in NetCDF format (http://edgar.jrc.ec.europa.eu/htap_v2/index.php?SECURE=123).

### 2.1.2 ECLIPSE V5a

ECLIPSE v5a (hereafter "ECL") anthropogenic emissions data is created by the Greenhouse gas-Air pollution Interactions and Synergies (GAINS) model, which contains information on the sources of emissions, environmental policies and mitigation efforts and opportunities for approximately 160 countries (Stohl, et al., 2015). The emission data have been rigorously evaluated through comparisons

with multiple ground-based and satellite observational data sets from Europe, Asia and the Arctic, with improvements for Arctic aerosols, when compared to previous studies (Stohl, et al., 2015). The emissions used are for the year 2010 at a resolution of 0.5° x 0.5°. Shipping emissions are available at a 1° x 1° resolution. ECL provides emissions for $SO_2$, $NO_x$, $NH_3$, NMVOC, BC, OC, $PM_{2.5}$, $PM_{10}$, CO and $CH_4$. split into different anthropogenic sectors (agricultural waste burning, residential, energy, industry,

transport, waste, and shipping).

### 2.1.3 Anthropogenic Emission Comparisons for Western Siberian Domain.

Comparisons between the two anthropogenic emission inventories for $NO_x$ show larger emissions in the

20 ECL inventory for Western Siberia (Fig. 2). $NO_x$ emissions within the domain are dominated by the

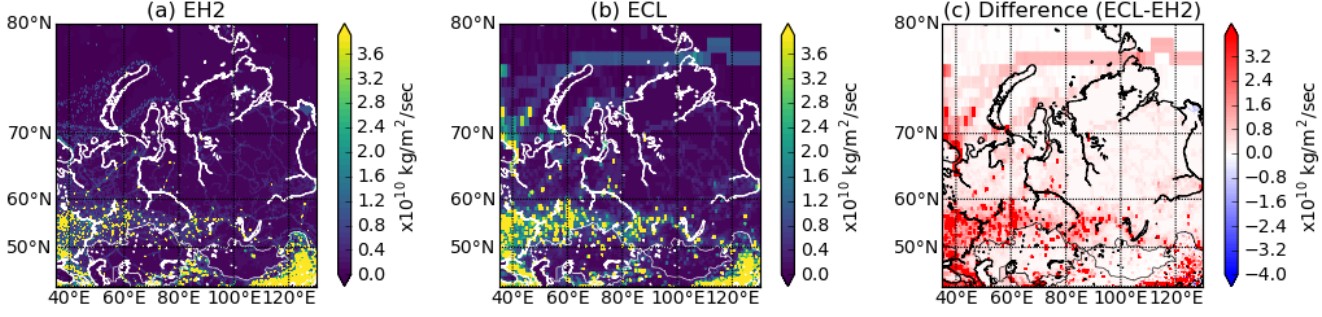

**Figure 2 –**Spatial distribution of anthropogenic emissions according to EDGAR HTAP v2.2 (panel (a)) and ECLIPSE v5a (panel (b)) inventories. Difference between the 2 inventories is shown in panel (c) (ECLIPSE v5a – EDGAR HTAP v2.2).



Transport and Energy sectors, which together contribute 75% of emissions for EH2, and 82% for ECL respectively (Table 1). For both emission inventories the largest sector contribution is from transport, which accounts for 41% of total EH2 emissions and 48% of total ECL emissions. Figure 2 shows that despite larger magnitude of emissions in ECL, with the largest difference seen over the urban regions

within the domain, the spatial patterns of total emissions are similar in both inventories. Differences are also seen in the shipping emissions, with large emissions north of Murmansk in the ECL inventory, which is not seen to the same extent in EH2 emissions. ECL attempts to better account for point source emissions associated with gas flaring above 60°N, which can be seen between 60°E – 80°E.

**Table 1 - Total NOₓ emissions (kilotons per month) for the study domain from EH2 and ECL anthropogenic emission inventories, and soil NOₓ contribution from GEIA. Contributions from energy and transport sectors shown for each inventory.**

| | EH2 Total | EH2 Energy | EH2 Transport | ECL Total | ECL Energy | ECL Transport | GEIA Soil NOₓ |
|---|---|---|---|---|---|---|---|
| **April** | 987.6 | 356.4 | 369.0 | 1067.8 | 377.8 | 473.7 | 9.5 |
| **May** | 915.5 | 306.0 | 375.2 | 989.8 | 324.4 | 481.7 | 51.4 |
| **June** | 911.3 | 307.8 | 374.1 | 985.3 | 326.3 | 480.3 | 71.9 |
| **July** | 870.4 | 297.1 | 367.8 | 941.1 | 315.0 | 472.2 | 84.1 |
| **August** | 864.4 | 294.0 | 368.7 | 934.6 | 311.7 | 473.3 | 88.9 |
| **Total** | **4549.2** | **1561.3** | **1854.9** | **4918.6** | **1655.2** | **2381.2** | **305.7** |

## 2.2 Anthropogenic Soil NOₓ Emissions

Past studies have highlighted potential missing sources of anthropogenic soil NOₓ emissions in current inventories (Ganzeveld et al., 2010; Jaeglé et al., 2005; Visser et al., 2019). In particular it is suggested that during the summer in northern mid-latitude regions, soil NOₓ emissions can contribute up to half

those from fossil fuel combustion, which could have important impacts upon background ozone concentrations (Jaeglé et al., 2005). The missing source is attributed to strong levels of fertilized agricultural soils, which are not well represented in current global or regional models. Estimates of global soil NOₓ emissions have been undertaken through different methodologies which include using top-down emission estimates (Vinken et al., 2014); scaling based upon multiple field measurement campaigns

(Davidson and Kingerlee, 1997); and using an empirical model (Steinkamp and Lawrence, 2011; Yienger

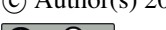



and Levy, 1995). Despite this, global soil NO$_x$ estimates vary significantly (9-27 Tg per year) (Oikawa et al., 2015). Agricultural NO$_x$ emissions are available within the EH2 inventory but missing in ECL, therefore we supplement all ECL simulations with additional anthropogenic soil NO$_x$. These are from the GEIA global soil NO$_x$ anthropogenic emissions, and distributed spatially according to the Yienger and Levy (1995) empirical model. The contributions per month from the anthropogenic Soil NO$_x$ emissions from this dataset to Western Siberia are shown in Table 1.

### 2.3 Observational Data

#### 2.3.1 Surface Sites

Tomsk observations are from the Fonovaya Observatory, which is an Institute of Atmospheric Optics (IAO) observational site, part of the Russian Academy of Sciences (RAS) Siberian Branch (Antonovich et al., 2018). This is located 60 km to the west of Tomsk (approx. 57°N, 85°E) in a rural, boreal location. Hourly ozone measurements are available at the surface from 2010 – 2011 (Davydov et al., 2018). These measurements are taken using an OPTEC 3.02-P chemiluminescence analyser at 10 m on an observational mast outside of the Observatory. Near real-time graphical representation of the data is available at http://lop.iao.ru/EN/.

The Zotino Tall Tower Observatory (ZOTTO) is situated in central Siberia (61°N, 89°E). The tower is 304 m in height, with 6 measurement platforms at 4, 52, 92, 158, 227 and 301m for meteorological variables, and 2 air sampling inlets positioned at 30 and 301m for ozone and NO$_x$ measurements carried out by Dasibi 1008AH-type and Thermo Electron Model 42C-TL gas analysers, respectively (Moiseenko et al., 2019). At present, human impacts on the local air quality are minimal due to the low population density of the area. These observations are therefore useful in evaluating the background atmospheric composition in the central Siberian region. In this study we use hourly ozone measurements taken from 30 m.





The Tiksi Observatory (71.36°N, 128.53°E) is located at the mouth of the Lena River, in remote northern Russia. It is situated in a region far from any major sources of anthropogenic pollution, other than the town of Tiksi (5000 population) which is 5 km northeast of the observatory. This location offers an opportunity to gain observations at high latitudes in a near pristine environment. At present, the

observatory is run in collaboration with NOAA (National Oceanic and Atmospheric Administration), the Tiksi Data Centre at the Arctic and Antarctic Research Institute in St Petersburg, Russia, which is responsible for the collection and distribution of the data, the Yakutian Service for Hydrometeorology and Environmental Monitoring, and the FMI (Finnish Meteorological Institute) (Asmi et al., 2016; Uttal et al., 2016). For this study, we use hourly $O_3$ concentrations measured with a Thermo Scientific Model

19*i* analyser, which are available for 2011 ([https://www.esrl.noaa.gov/gmd/dv/iadv/](https://www.esrl.noaa.gov/gmd/dv/iadv/)).

### 2.3.2   Ozone Monitoring Instrument satellite data

We make use of satellite data from the Dutch OMI (Ozone Monitoring Instrument) for tropospheric $NO_2$

(DOMINO v2.0), on-board NASA's polar orbiting Aura satellite, launched in 2004 (Boersma et al., 2011; Vinken et al., 2014). OMI retrievals of trace gases are through the Differential Optical Absorption Spectroscopy (DOAS) method, which involves using the on-board spectrometer to make UV-visible measurements. This provides tropospheric column $NO_2$ through first calculating the slant columns, which is the quantity of $NO_2$ along the whole photon path through the atmosphere to the instrument (Vinken et

al., 2014). Using a tropospheric air mass factor (AMF), tropospheric vertical column density (VCD) of $NO_2$ can be retrieved, which is mapped onto a 0.25° x 0.25° grid. This data was provided on a daily temporal scale and has been averaged into monthly means (April-August) to provide reliable spatial coverage at high latitudes. To allow for a direct comparison of OMI with modelled column $NO_2$, averaging kernels are applied to model fields (Pope et al., 2015), to account for OMI vertical sensitivity

varying through the tropospheric profile. The averaging kernel provides a relative sensitivity of the satellite instrument to the abundance of species of interest at different vertical points within the column (Herron-Thorpe et al., 2010). We apply averaging kernels to WRF-Chem that are provided as a column vector alongside the total column retrieval for $NO_2$ from the DOMINO product.



All column comparisons presented in this work are undertaken at 0.5° x 0.5° resolution and limited to south of 65°N latitude. This latitude is chosen as a cut-off for the comparisons, since satellite retrieval uncertainty increases at higher latitudes, for solar zenith angle greater than 70°. Furthermore, ± 65° is the latitudinal range used to map global NO2 VCD when using the DOAS retrieval method (Bucsela et al., 2006).

## 2.4. Model Simulations

We use the Weather Research and Forecasting model coupled with chemistry (WRF-Chem) version 3.7.1 (Grell et al., 2005) to simulate tropospheric chemical and aerosol composition over Western Siberia. WRF-Chem is a fully coupled online model, in which atmospheric chemistry and meteorological components are fully consistent, using the same transport scheme, time step, advection, and physics schemes. The model domain (Fig. 1) has a 30 km x 30 km horizontal resolution, in a 140 x 140 grid. There are 32 vertical levels, with the model top at 10 hPa, and the model uses terrain following hydrostatic pressure coordinates. Model gas phase chemistry is simulated using the Model of Ozone and Related Chemical Tracers v4 (MOZART-4; Emmons et al., 2010), whilst the model aerosol scheme is the 4-bin Model for Simulating Aerosol Interactions and Chemistry (MOSAIC; Zaveri et al., 2008), using chemistry option 201, but with updates to aromatic photochemistry, biogenic hydrocarbons, and further species which are important for regional air quality (Hodzic and Jimenez, 2011; Knote et al., 2014). Biogenic emissions are calculated online using the Model of Emissions of Gases and Aerosols from Nature (MEGAN; Guenther et al., 2006). Biomass Burning emissions are from the Fire Inventory from NCAR (FINN) for 2011 (Wiedinmyer et al., 2010). The dry deposition scheme used in this model setup is the Wesley Scheme (Wesley, 1989), and we use the modified International Geosphere-Biosphere Programme (IGBP) Moderate Resolution Imaging Spectroradiometer (MODIS) Noah land surface scheme (Ek, 2003), which has 20 land surface types. For more information on model setup please refer to Supplementary Table 1.



Model simulations are conducted between April and August 2011, with a spin-up period of 2 weeks preceding this. This simulation length is chosen at it represents the optimum period of time for valid OMI satellite comparisons at the latitudes of interest, and the year 2011 has good surface observation data availability within the domain. WRF-Chem has successfully been used at high latitudes previously to

investigate air quality issues (Marelle et al., 2015; Marelle et al., 2017 Raut et., 2017; Stohl, 2007; Thomas et al., 2013), with model output being compared to both flight campaigns and ground observations. Three separate 5-month sensitivity simulations are conducted, within each of which one of three different emission sources are removed: biomass burning emissions (fires_off simulation), anthropogenic transport emissions (trans_off simulation), and anthropogenic energy emissions (ene_off simulation). Model

evaluation with observations is presented in Sections 3.1 and 3.2, for two control simulations, each using one of the two anthropogenic emissions inventories. The subsequent sensitivity simulations in Section 3.3 use the optimal inventory for the domain based on this evaluation.



# 3    Results

## 3.1. OMI – model Comparisons

For both anthropogenic emission inventories (EH2 and ECL), an overall negative bias is seen in WRF-Chem tropospheric column $NO_2$ when compared with OMI satellite observations (Fig. 3). The greatest negative bias is during June for both anthropogenic emission inventories (Fig. 3h & 3m). During June, July and August there is a statistically significant negative bias in the southwestern section of the domain using both anthropogenic emission inventories (highlighted by hatching in Figure 3). This significance is most prominent during the June and July months, particularly over urban regions in the EH2 simulation (Fig. 3h). For large parts of the domain that are located further from large anthropogenic sources, there is better agreement between the observed and modelled column $NO_2$ values.

Over urban regions with large emissions south of 60°N, OMI tropospheric column $NO_2$ distributions show values exceeding $2 \times 10^{15}$ molec $cm^{-2}$ (Fig. 3a-e), with some variability across the 5-month period. During April, many urban regions display a positive model bias, particularly when using the ECL anthropogenic emissions (Fig. 3k). Tropospheric column $NO_2$ biases greater than $1 \times 10^{15}$ molec $cm^{-2}$ are seen over the major cities within the north-western section of the domain, for example in Kazan, Perm, Yekaterinburg and Ufa, whilst also showing positive biases over the cities more centrally located, such as Tomsk and Novosibirsk.



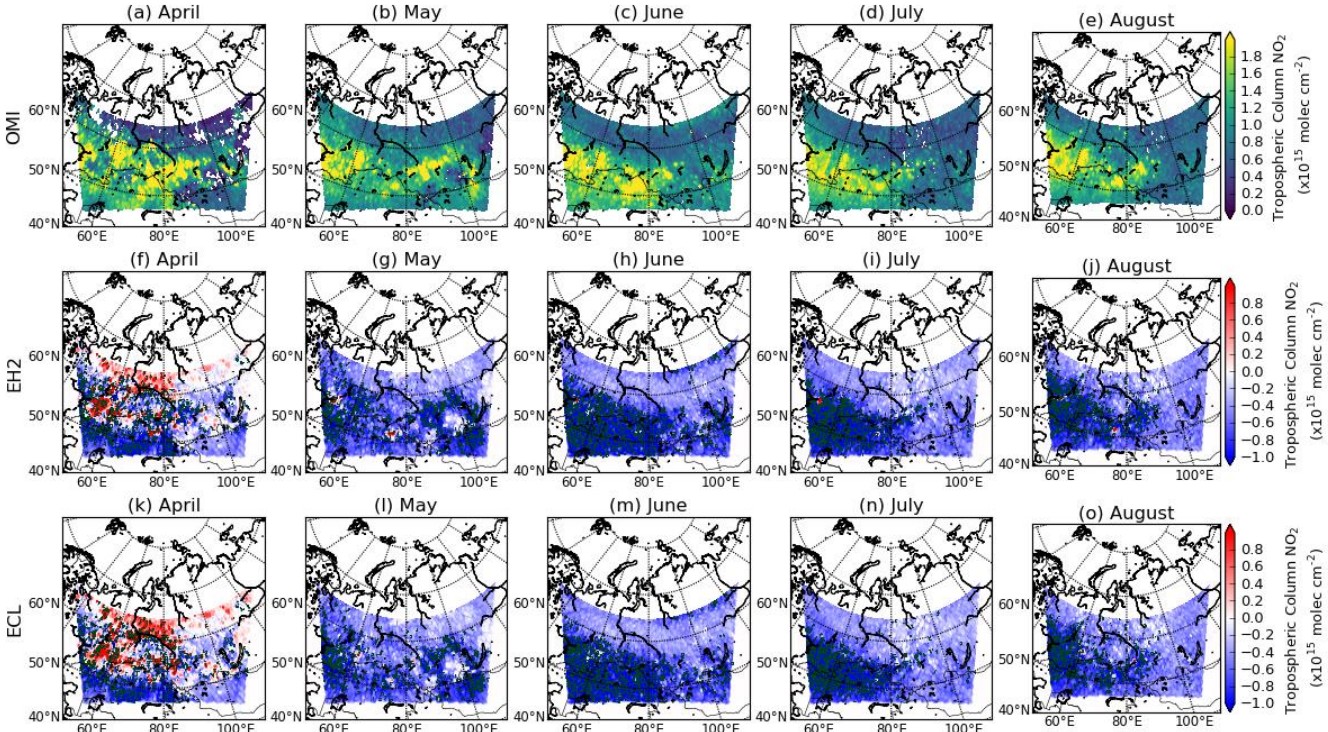

**Figure 1 –** Observed and model-observed tropospheric column NO₂. Panels a-e show mean OMI tropospheric column NO₂ for April-August. Panels f-j show WRF-Chem mean bias (model – satellite) using the EDGAR HTAP v2.2 anthropogenic emission inventory for April-August. Panels k-o show WRF-Chem mean bias using the ECLIPSE v5a anthropogenic emission inventory for April-August. Results not shown <65°, due to satellite retrieval uncertainty increasing at high latitudes, associated with a large solar zenith angle. Hatching shows where modelled values of tropospheric column NO₂ are outside of the satellite uncertainty range

For all 5 months the WRF-Chem simulations using ECL anthropogenic emissions provide better agreement with observations for tropospheric column NO₂ over Western Siberia (Fig. 4). Despite this, negative biases persist across the whole simulated period with both anthropogenic emission inventories. In particular, negative biases are marked during June and July using either anthropogenic emission inventory, reflected in the regression slope values for ECL during June (slope = 0.2), and EH2 during July (slope = 0.1). However, better correlation coefficients are produced using ECL anthropogenic emissions during July (r = 0.74) and August (r = 0.74), and EH2 anthropogenic emissions during August



(r = 0.55), which suggests spatial patterns in NO₂ sources are well simulated, especially in ECL but may be underestimated. Model-observation correlations are poorest in April for both inventories (EH2 r = 0.36; ECL r = 0.40), with a large degree of scatter compared with other months. This is consistent with the model displaying both an overestimation and underestimation of NO₂ in urban and background regions, respectively, with low bias on average (see Fig. 3f, k).

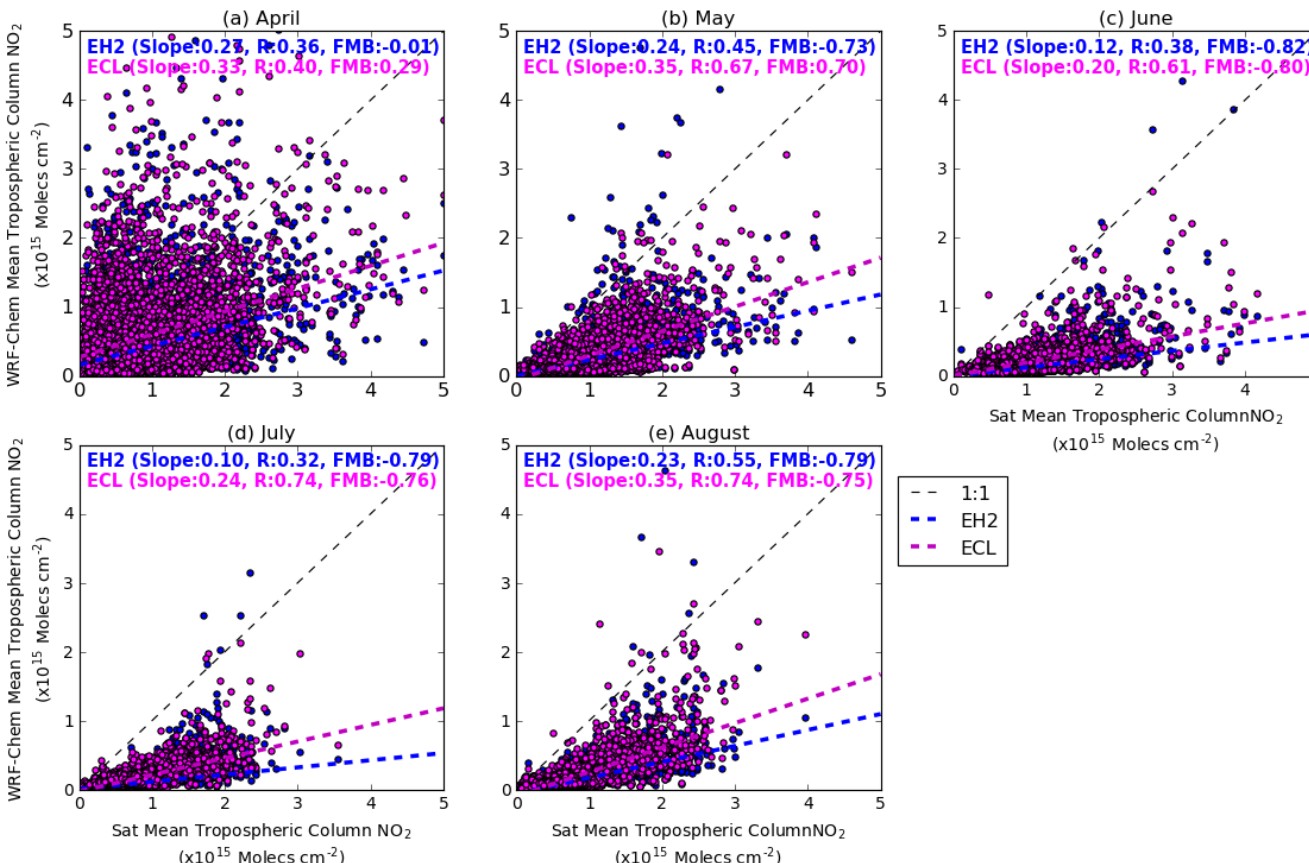

**Figure 2** – OMI tropospheric column NO₂ against WRF-Chem simulations using ECL (magenta) and EH2 (blue) anthropogenic emissions. Panel (a) shows April mean; Panel (b) shows May mean; Panel (c) shows June mean; Panel (d) shows July mean; Panel (e) shows August mean. All plots show total domain below 65°N. Slope, correlation coefficient (R) and fractional mean bias (FMB) are shown for all plots.



Major cities located within western Siberia show smaller fractional mean biases for tropospheric column $NO_2$ when using the ECL anthropogenic emission inventory across the whole study period (Fig. 5). This is especiically the case over Novosibirsk, Novokuznetsk and Tomsk in the centre of the domain, where the mean fractional bias is larger for almost all months when using the EH2 anthropogenic emissions,

Novosibirsk in August being the exception. The transport sector is the dominant source for $NO_x$ in ECL and EH2 over Novosibirisk and Tomsk, whilst in Novokuznetsk it is the industrial sector (EH2) and energy sector (ECL). Despite the different dominant sector over Novokuznetsk, a mean negative bias is seen across all 5 months using both inventories.

The same overall pattern is replicated at the other major cities within the western section of the domain, where we see predominantly lower fractional mean biases using the ECL anthropogenic emissions at nearly all cities (Kazan and Pavlodar being the only exceptions). In both of the anthropogenic emission inventories the cities in this western section of the domain are dominated by $NO_x$ emissions from the transport sector, with Yekaterinburg in the EH2 inventory the only city with a different anthropogenic

sector as its main $NO_x$ source sector (Industry). The model bias could therefore suggest a potential underestimation of $NO_x$ emissions in the transport sectors of both anthropogenic inventories over urban areas.





**Figure 5** –Fractional mean bias of monthly simulated tropospheric column NO$_2$ for major cities (population >100,000) within Western Siberia when compared with OMI values. Panel (a) shows results using the EH2 anthropogenic emission inventory. Panel (b) shows results using the ECL anthropogenic emission inventory.



## 3.2 Surface Observation – model Comparisons

We evaluate surface ozone in WRF-Chem over the same April-August 2011 period using observations from Tomsk, ZOTTO, and Tiksi, described in Section 2.3.1 (Fig. 6). Our model simulations suggest that tropospheric ozone in these locations is relatively insensitive to the choice of anthropogenic emissions inventory. This is particularly the case at the 2 background sites of ZOTTO (EH2 FMB: 0.05; ECL FMB: 0.12 and Tiksi (EH2 FMB: 1.32; ECL FMB: 1.32). This suggests that the effects of the differences between the two emission inventories are minimal in the background ozone concentrations. The greatest sensitivity to the differences between the anthropogenic emissions datasets is at Tomsk, where the model/observation correlation is improved using the EH2 emissions (EH2 R: 0.77; ECL R: 0.69), although with a slightly larger FMB (EH2 FMB: 0.37; ECL FMB: 0.35).

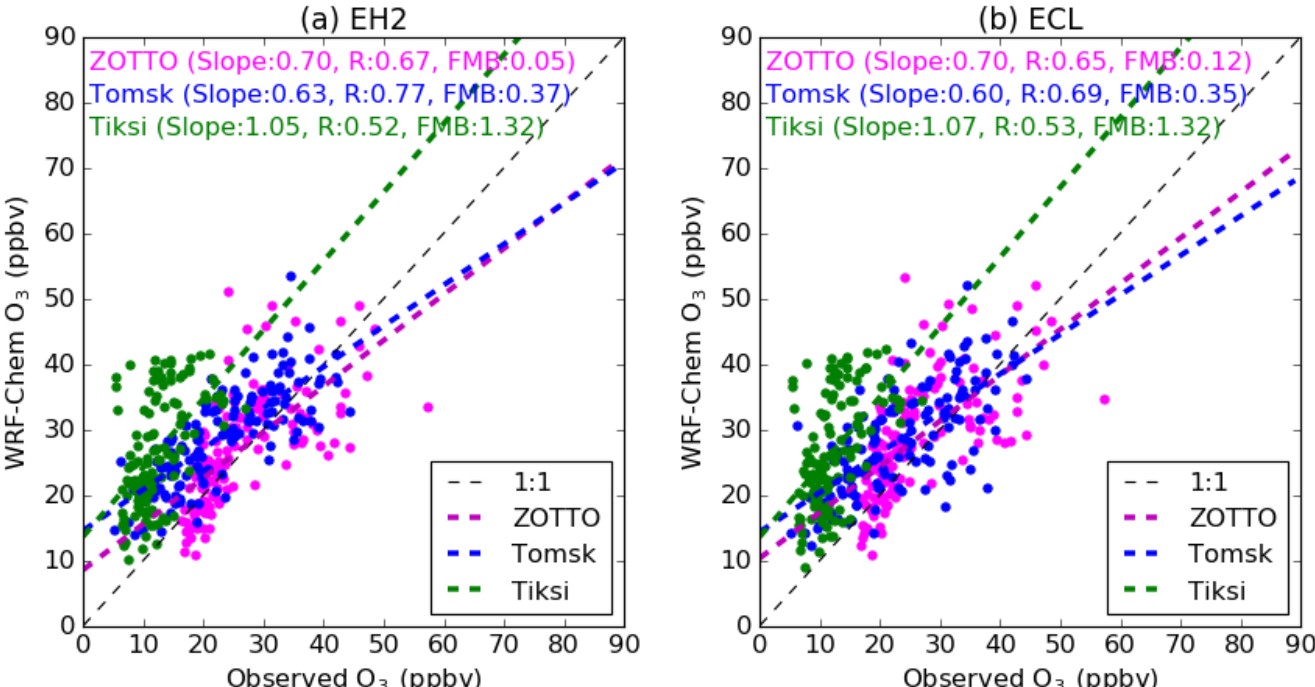

**Figure 6** – Daily mean surface ozone comparisons for three ground observation sites within the study domain: ZOTTO (magenta), Tomsk (blue) and Tiksi (green) for 01/04/11 – 31/08/11. Panel (a) shows WRF-Chem surface ozone using EH2 anthropogenic emissions; panel (b) shows WRF-Chem surface ozone using ECL anthropogenic emissions.

There is a consistent positive bias in modelled surface ozone values at all 3 observation sites, with the largest bias seen at Tiksi with both simulations displaying a fractional mean bias of 1.32. Both model simulations also show substantially lower fractional mean bias values for ZOTTO when compared to the other observation sites, EH2 (FMB=0.05) and ECL (FMB=0.12). Due to the largely boreal forest land surface cover at the ZOTTO observation site, this could suggest improved model performance over boreal forest regions, when compared to the Arctic tundra at Tiksi.

## 3.3 Sensitivity Studies

Based upon the results in Sections 3.1 and 3.2, from this point onwards we use model simulations with the ECL anthropogenic emissions to perform sensitivity simulations, since for tropospheric column $NO_2$



in particular, these emissions produced a smaller bias against observations averaged across the domain. The three sensitivity simulations are used to gain a better understanding of the impacts of transport (trans_off), energy (ene_off) and fire (fires_off) emissions. Transport and energy are chosen as the two dominant anthropogenic $NO_x$ emission sectors, representing 82% of total $NO_x$ emissions in the ECL

anthropogenic emission inventory for the modelled domain. From here forward, the simulation in which anthropogenic (ECL) and fire emissions are standard will be termed the Control.

### 3.3.1 NO₂ Source Contributions

Simulated surface $NO_2$ concentrations show enhancements in regions close to major anthropogenic emission sources, mainly urban regions south of 60°N, throughout the 5-month study period (Fig. 7 a-f). North of 60°N the major contribution is from the energy sector, due to significant gas flaring activity associated with natural gas extraction. Enhanced background concentrations of surface $NO_2$ are simulated

at high latitudes during April with mean surface $NO_2$ concentrations north of 60°N of 0.4 ppbv, which is more than 50% greater than any other month during the study. This is associated with a longer $NO_2$ lifetime, and late-springtime poleward advection of air from regions south of 60°N.

In the fires_off sensitivity simulation (Fig. 7 f-j) there is a small reduction in $NO_2$ concentrations,

including background regions not in close proximity to fire source regions, across all 5 monthly periods. The main feature of the fires_off simulation is the impact of a major fire event during June, which can be seen to the east of the Ob Valley region (approx. 58°N, 100°E) (Fig. 7h), resulting in a maximum lowering of surface $NO_2$ by ~10% for the area in close proximity to the fire.

As expected, given their relative source sizes, $NO_2$ concentrations in Western Siberia are most sensitive to anthropogenic emissions relating to transport and energy activities (Fig. 7k-t), rather than those associated with fires. This is particularly the case during April for both anthropogenic sensitivity simulations, when simulated $NO_2$ reductions are more widespread due to the longer $NO_2$ lifetime. Across the whole domain relative to the control simulation during April, we see a reduction of 0.4 ppbv in both





the trans_off and ene_off simulations. North of 60°N, compared to the control during April there is a greater NO$_2$ reduction in the trans_off simulation (-0.2 ppbv) compared with the ene_off simulation (-0.1 ppbv). Transport sector emissions are the largest source of surface NO$_2$ during the 5-month simulation (Fig. 7k-o). A widespread reduction of surface NO$_2$ is simulated in the trans_off simulation south of 60°N,

5    both close to the urban source regions and in between cities, associated with on-road transport emissions. We also see reductions in surface NO$_2$ in the ene_off simulation confined to the major urban regions (Fig. 7p-t), which is likely due to emissions being point sources of high emissions associated with energy production facilities. North of 60°N the influence of high latitude gas flaring emissions is evident, which have greatest impact on NO$_2$ in August (Fig. 7t). Reductions in the abundance of peroxyacetyl nitrate

10   (PAN) sourced from lower latitude NO$_2$ likely also play a role in reducing high latitude NO$_2$ abundance in the emission perturbation simulations (see Fig S2).

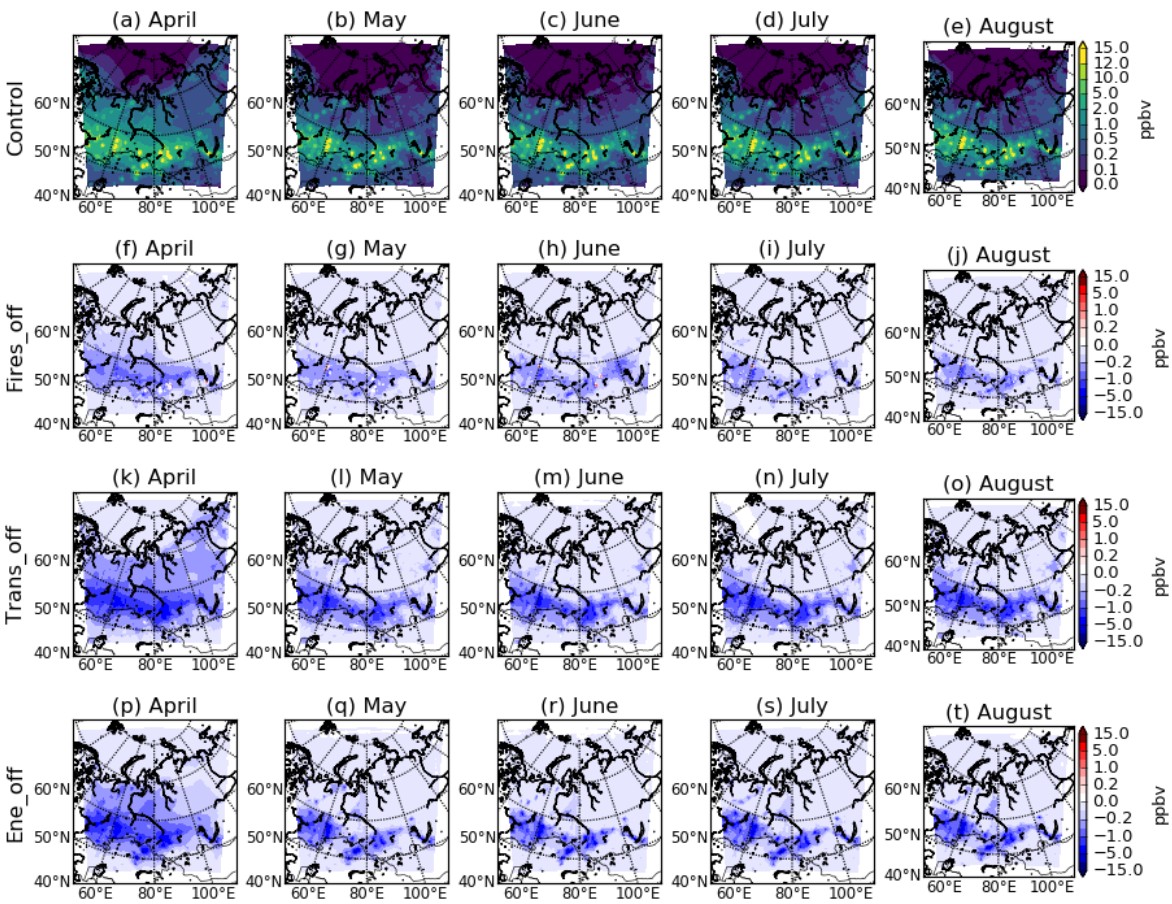

**Figure 7–** Simulated control and sensitivity run changes in surface NO2 concentrations. Panels (a)–(e) show monthly means of WRF-Chem surface NO2 for April-August. Panels (f)-(j) show monthly means of WRF-Chem Surface NO2 with all fire emissions switched off in domain (fires_off simulation) minus control simulation for April-August. Panels (k)-(o) show monthly means of WRF-Chem Surface NO2 with all transport emissions switched off in domain (trans_off) minus control simulation for April-August. Panels (p)-(t) show monthly means of WRF-Chem Surface NO2 with all energy emissions switched off in domain (ene_off) minus control simulation for April-August.




### 3.3.2 Ozone Source Contributions

Surface ozone concentrations in Western Siberia are larger during April (mean=35.7 ppbv) (Fig. 8a) and May (mean=34.6 ppbv) (Fig. 8b), coinciding with the well-characterised springtime peak of in Arctic surface ozone (Quinn et al., 2008; Stohl et al., 2007). This has been attributed to poleward import of ozone precursors, or an increase in stratospheric downwelling, which is more frequent during springtime at high latitudes (Berchet et al., 2013). In July and August (Fig. 8d-e) a surface ozone gradient from north to south begins to emerge and is strongest during August (Fig. 8e). This results from lower simulated ozone concentrations over the Arctic, where the mean surface ozone concentration above 60°N is 20.0 ppbv, whilst below 60°N it is 32.3 ppbv. These low levels of modelled surface ozone seen at high latitudes occur as wind directions change to a northerly direction during summer, limiting the import of ozone precursors from lower latitudes into the Arctic. During June-August (Fig. 8d-f), largest surface ozone concentrations occur over the areas of significant precursor emissions, where monthly surface ozone averages across the summer exceed 35 ppbv. During June (Fig. 8c) concentrations exceeding 45 ppbv are simulated in the region to the east of the Ob valley, which is associated with a major fire event during this month.

Surface ozone is most sensitive to anthropogenic emissions, particularly those from the transport sector (Fig 9). This is the case for both above and below 60°N, where we see the maximum differences relative to the control simulation for the transport off simulation occur in May north of 60°N (-0.9 ppbv) and in July (-2.5 ppbv) south of 60°N. Across the entire 5-month period for the total domain, widespread reductions in surface ozone concentrations are seen in both the trans_off (Fig. 8k-o) and ene_off (Fig. 8p-t) simulations. However, within the ene_off simulations an increase in ozone is simulated over urban regions with high anthropogenic emissions, due to a decrease in the loss of ozone via $NO + O_3$ where $NO_x$ emissions are reduced. In the fires_off simulation there is a small reduction over a large area in surface ozone south of 60°N in April (Fig. 8f) and May (Fig. 8g), whereas in June (Fig. 8h) we see a significant reduction of surface ozone to the east of the Ob valley, which is the location of a major biomass burning event.



Anthropogenic emissions from the energy and transport sectors sourced from within the domain contribute more to surface ozone north of 60°N than fire emissions for all months, other than May and June (Fig. 9), with surface ozone sourced from fires predominantly confined to south of 60°N. In the high fire month of June, we see the greatest influence of fires on surface ozone north of 60°N for the entire study period, but the difference compared to the control simulation is less than 1 ppbv. During April north of 60°N large contributions to surface ozone from energy emissions are seen, likely due to emissions associated with high latitude oil and gas extraction within the domain. This contribution is enhanced due to poleward movement of air which occurs during late springtime (Stohl, 2007) (Fig. S1). There is a shift in wind direction north of 60°N during May to a more northerly flow bringing in cleaner Arctic air, which leads to efficient southward export of the energy sourced ozone at high latitudes, evident as an increase in surface ozone south of 60°N during May (Fig. 9).

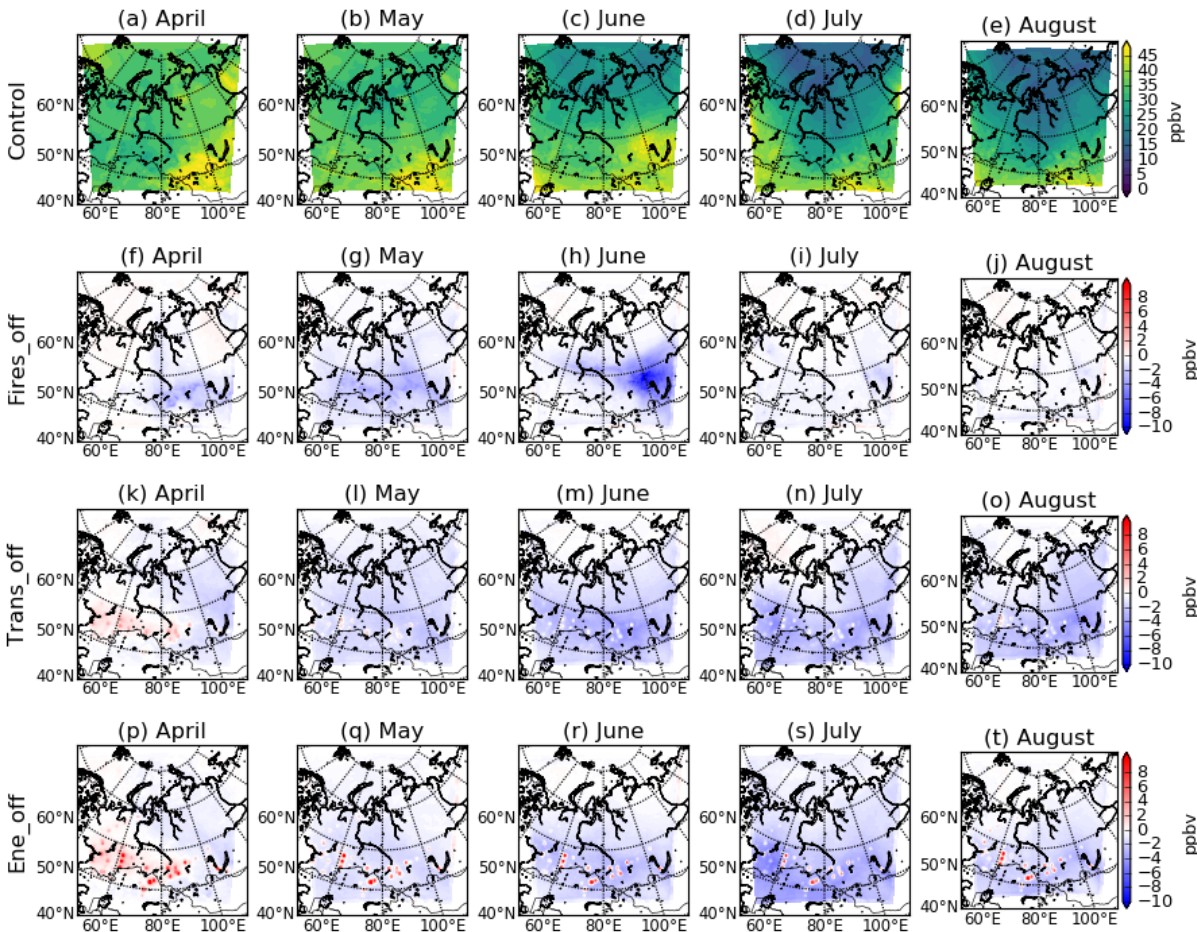

**Figure 8** – Simulated control and sensitivity run changes in surface ozone concentrations. Panels (a)–(e) show monthly means of WRF-Chem surface ozone for April-August. Panels (f)-(j) show monthly means of WRF-Chem Surface ozone with all fire emissions switched off in domain (fires_off simulation) minus control simulation for April-August. Panels (k)-(o) show monthly means of WRF-Chem Surface ozone with all transport emissions switched off in domain (trans_off) minus control simulation for April-August. Panels (p)-(t) show monthly means of WRF-Chem Surface ozone with all energy emissions switched off in domain (ene_off) minus control simulation for April-August.





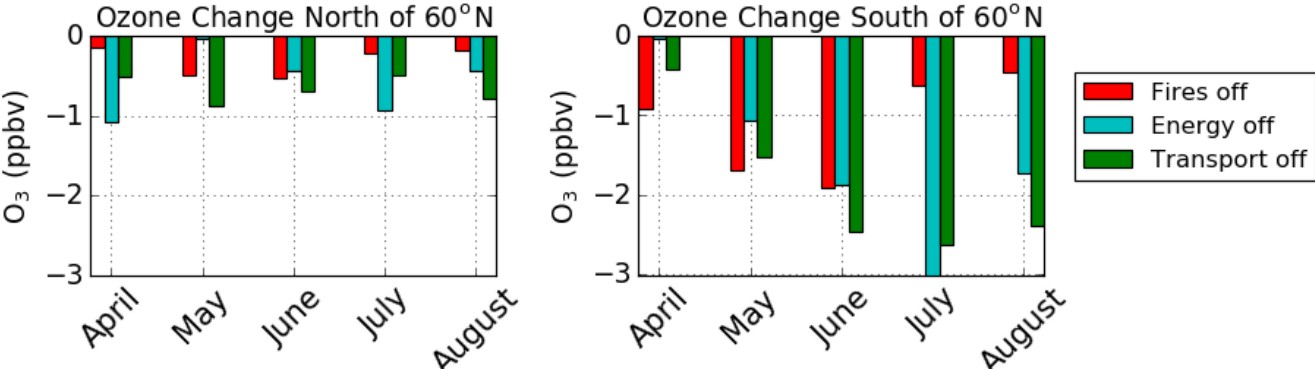

**Figure 9** – Surface ozone change relative to control simulation for the section of the domain north of 60°N (left panel) and south of 60°N (right panel) for the 3 sensitivity simulations, fires_off (red), ene_off (blue) and trans_off (green).





## 3.4 Ozone Dry Deposition

To investigate the impact of vegetation as a surface sink of ozone in Western Siberia, we analyse ozone dry deposition fluxes output from the WRF-Chem simulations (Fig. 10). These fluxes are partitioned across each of the 20 land surface types from the IGBP MODIS Noah land surface scheme used in the model. We group similar land surface types to provide total fluxes over 8 land cover categories (Fig. 11). Maximum ozone deposition to the surface occurs during the summer months of June, July & August (Fig. 10c-e), which coincides with the summer peak in photosynthesis in vegetation (Karlsson et al., 2007; Stjernberg et al., 2012). Dry deposition fluxes are lower during April (Fig. 10a) and May (Fig. 10b), coincident with the period of highest concentrations for modelled surface ozone (Fig. 8a-b).

Ozone dry deposition flux is most sensitive to anthropogenic ozone precursor emissions, especially during June, July and August (Fig 10). The reduction in dry deposition flux in the anthropogenic perturbation simulations is greater south of 60°N during this period but extends north of 60°N in July and August, for both the trans_off and ene_off simulations. This is despite low levels of surface ozone at high latitudes during these months (Fig. 8d-e). This is likely due to enhanced photosynthesis activity and stomatal conductance during the summertime, which leads to this period being the most active months in terms of ozone deposition flux.



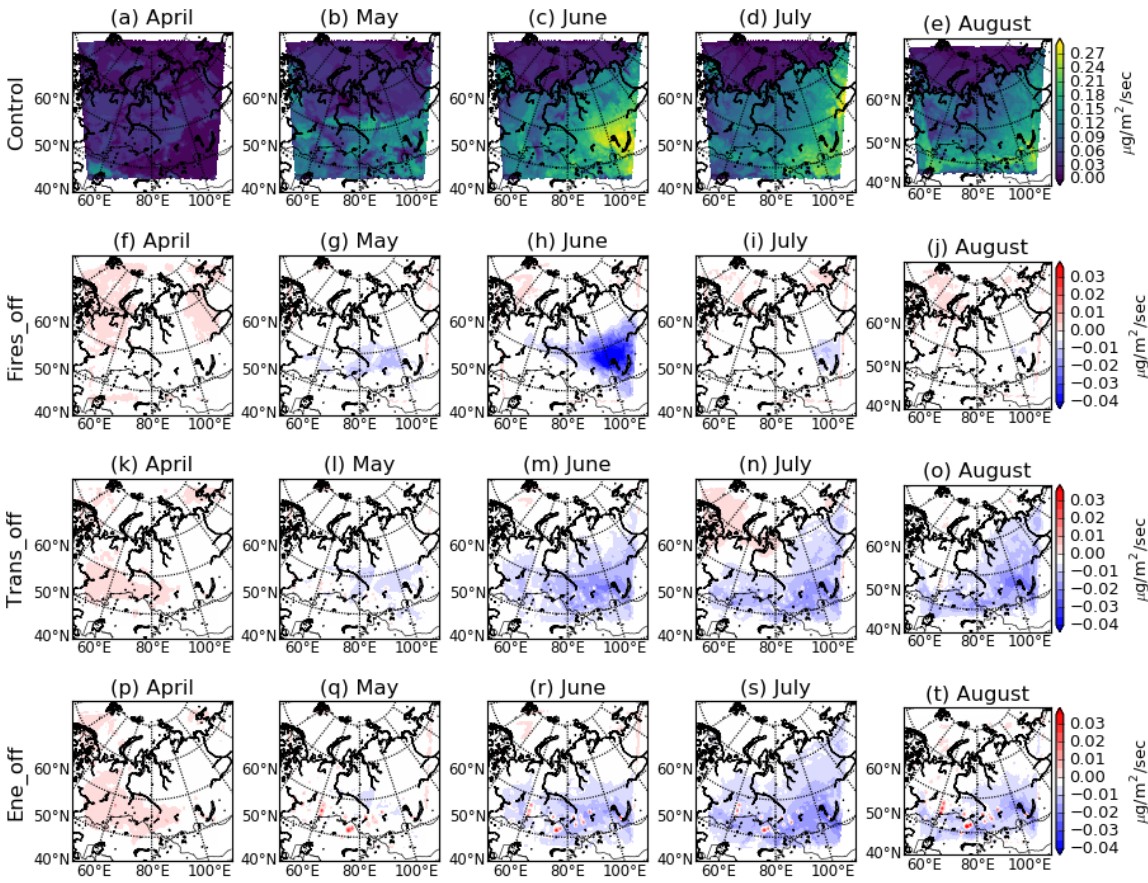

**Figure 10 –** Simulated control and sensitivity run changes in surface ozone dry deposition fluxes. Panels (a)–(e) show monthly means of WRF-Chem surface ozone deposition flux for April-August. Panels (f)-(j) show monthly means of WRF-Chem Surface ozone flux with all fire emissions switched off in domain (fires_off simulation) minus control simulation for April-August. Panels (k)-(o) show monthly means of WRF-Chem Surface ozone with all transport emissions switched off in domain (trans_off) minus control simulation for April-August. Panels (p)-(t) show monthly means of WRF-Chem Surface ozone with all energy emissions switched off in domain (ene_off) minus control simulation for April-August.





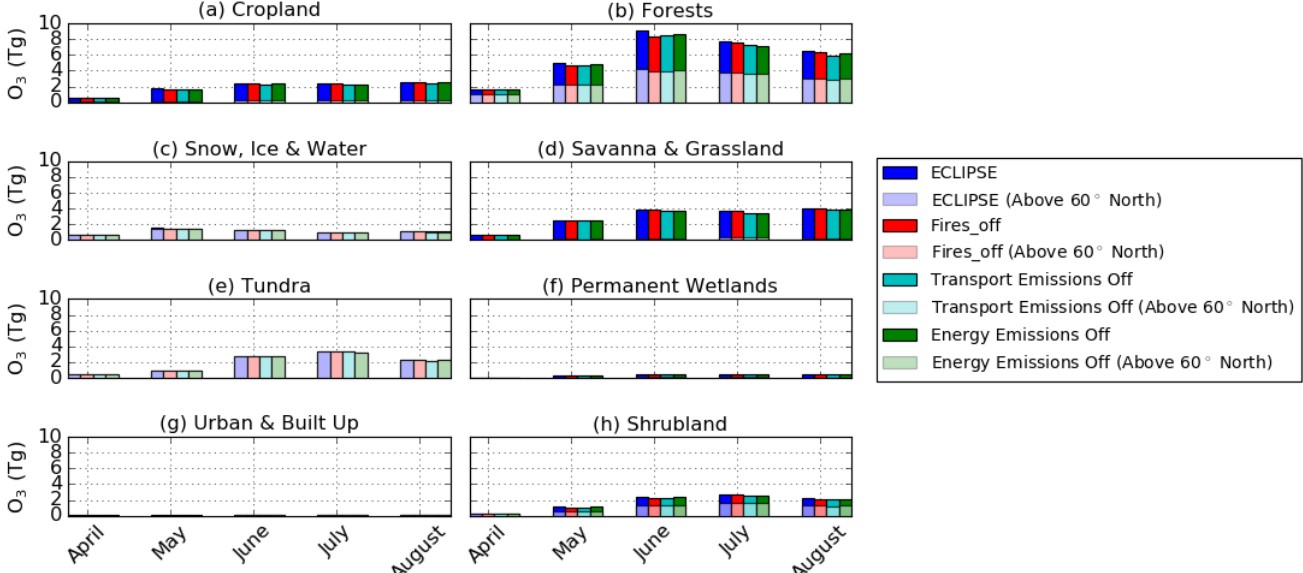

**Figure 11** – Quantity of ozone deposited to modified IGBP MODIS NOAH land surface cover categories per month for total domain (solid bars) and for the section of the domain above 60°N (pale bars).

The largest deposition sink for ozone in the model domain is to forest vegetation, averaging 6.0 Tg of ozone deposition per month in the control simulation, peaking at 9.1 Tg of ozone deposition during June (Fig. 11b). Forest covers 29% of the domain, spanning large areas both north and south 60°N. For the total domain, "cropland & vegetation" and "savanna & grassland" surface types account for an average of 1.9 and 3.0 Tg/month of ozone loss, respectively. North of 60°N, forest and tundra are the dominant sinks, which account for 65% of dry deposition flux, and 77% of the terrestrial surface cover at these latitudes.

Ozone deposition flux responds most in the trans_off and ene_off simulations, in particular during July and August. Deposited ozone reduces by 8% over forests when anthropogenic energy emissions are removed during July, and by 9% when anthropogenic transport emissions are removed during August. The impact of fires on ozone dry deposition within the domain is small compared to anthropogenic emissions, and is negligible north of 60°N. The greatest impact of the fires_off simulation on ozone



deposition occurs during May and June, with the largest percentage change for forest land cover (May: 4%, June: 2%).





## 4    Discussion

There are limited studies comparing WRF-Chem and OMI tropospheric column NO$_2$, especially at high latitudes, but this technique has been shown in previous studies to be an effective regional model
evaluation tool (Han et al., 2011; Herron-Thorpe et al., 2010; Pope et al., 2015). Our results are consistent with previous work using OMI tropospheric NO$_2$ columns to evaluate an ensemble of regional models at similar latitudes over Europe (Huijnen et al., 2010). This showed low model biases for tropospheric column NO$_2$, which were greatest in magnitude during summertime in background regions, with ensemble mean column NO$_2$ values up to 50% lower than OMI. Furthermore, it was shown that greatest
spread between models occurred during the summer, with model underestimation ranging from 40-60% depending on the region. A positive bias in the DOMINO v1 product has been identified in previous studies of up to 40% in summer (Hains et al., 2010; Huijnen et al., 2010; Lamsal et al., 2010; Zhou et al., 2009), attributed to errors in the a priori NO$_2$ profile, air mass factors and albedo. These errors were improved in the DOMINO v2 product (Boersma et al., 2011), which we use here. This improved product
is shown to lower summertime satellite positive biases of tropospheric column NO$_2$ relative to retrievals using the previous version of the DOMINO product, and we therefore expect these retrieval issues to play less of a role in explaining our negative model bias during summer. We see widespread negative bias in WRF-Chem tropospheric column NO$_2$ when compared to the satellite measurements, especially during June (ECL FMB = -0.80) across the background regions of western Siberia, despite improvements in the
DOMINO v2.0 retrieval algorithm. These biases could also result from an underestimation of emissions and/or model deficiencies in NO$_x$ chemistry, leading to underestimation of the NO$_2$ lifetime. Kanaya et al., (2014) compared 2007 - 2012 OMI tropospheric column NO$_2$ retrievals using the DOMINO v2 product with Multi-Axis Differential optical absorption spectroscopy (MAX-DOAS) observations from multiple sites in Asia and one in Russia. The Russian site was located in Zvenigorod (55°N, 37°E)
approximately 50 km to the west of Moscow, where they found very good agreement between ground observations and satellite observations of tropospheric column NO$_2$, especially during the summer period of 2011 and 2012. Although limited in spatial scope, this comparison lends some limited confidence to the reliability of the OMI observations for this region during summer and may further support our model-observation differences being a result of poor representation of NO$_2$ sources or sinks in the model.





However, it is important to note that it is difficult to completely rule out errors in the DOMINO v2 retrieval, since it has not been extensively evaluated for this region.

We include anthropogenic soil $NO_x$ emissions in our model, which have been shown in previous studies to be a potentially overlooked source of $NO_x$ (Oikawa et al., 2015; Visser et al., 2019). Visser et al., (2019) highlighted potential underestimates in anthropogenic soil $NO_x$ emissions from the MEGAN emissions model, which resulted in negative model biases against surface observations of $NO_2$ across eastern Europe. Implementation of satellite-constrained surface $NO_x$ emissions inferred from OMI tropospheric column $NO_2$, subsequently improved low model bias in their analysis. In our study we supplement our ECL simulations with anthropogenic soil $NO_x$ emissions from the GEIA emission inventory. However, we find little impact from including these emissions on our model bias.

Cities within the domain demonstrate varying tropospheric column $NO_2$ biases, with localised model overestimates during the springtime at some locations. Other studies have shown that despite significant model underestimations of background tropospheric column $NO_2$ when compared to satellite observations, model performance is improved over cities (Huijnen et al., 2010). $NO_2$ underestimation persists in the model over the majority of major urban regions in the domain, particularly outside of spring. Our results show that for all cities in the Ob valley where the dominant anthropogenic $NO_x$ sector is transport, an underestimation is simulated for almost every month, regardless of anthropogenic emission inventory. The only exceptions were at Ufa and Yekaterinburg during April. Evans et al., (2017) suggest that the transport sector has grown dramatically between 2000-2013 in Russia, with a doubling of passenger vehicles, and a 40% increase in truck ownership. This rise of on-road vehicles may not be well represented within Western Siberian transport emissions in ECL and EH2, as global inventories often do not have access to up-to-date country wide data (Kholod et al., 2016).

Simulated tropospheric column $NO_2$ in the region is sensitive to the anthropogenic emission inventory used, with the ECL inventory providing an improved $NO_2$ simulation when compared to EH2 and OMI. ECL has been extensively used in previous high latitude, regional modelling studies (Marelle et al., 2018;



Sand et al., 2015; Stohl et al., 2013). Our results support the view that the ECL dataset is more suitable over the Western Siberia region compared with EH2. The ECL anthropogenic emission inventory attempts to add detail in the Arctic compared to EH2, accounting for better quantification of direct and associated emissions from gas flaring, and also a better understanding of emission seasonality (Stohl et

al., 2015). Despite this we still see a significant widespread low bias over the region, especially from May-August. Good spatial correlation (R=0.61-0.74) between model and OMI observations despite the low bias during this period further supports the possibility of an underestimation in sources. Huang et al., (2014) attribute potential unreliable representation of Russian anthropogenic emissions within global inventories due to difficulties in accurate quantification of local emission factors, and incorrect locations

of point sources.

The sensitivity of modelled surface ozone concentrations to the differences in the two anthropogenic emission inventories is small. This may be due to 2 of the ground observation sites being located far from precursor source regions (Tiksi & ZOTTO). We see a small improvement to the FMB values when using

ECL (FMB = 0.35) compared to EH2 (FMB =0.37) at the Tomsk observation site, which is closer to anthropogenic precursor sources. In Fig. 6a & b we see a positive bias in modelled surface ozone at all sites for both anthropogenic emission inventories. Skorokhod et al., (2011) suggest that during the night-time in Siberia ozone destruction can occur under intense temperature inversions through surface deposition to snowless surfaces. However, WRF-Chem does not perform well when recreating high

latitude temperature inversions with strongly stable stratification periods, which are often of high importance for high latitude surface air quality (Mölders et al., 2011; Schmale et al., 2018).

At Tiksi we see a positive bias in modelled surface ozone across the study period, which could be associated with missing halogen chemistry at high latitudes within our model. During ice melt the release

of bromine can lead to ozone depletion events, causing ozone concentrations to go from background concentrations (~30ppbv) to concentrations lower than 5ppbv within days (Cao et al., 2016; Zhao et al., 2016). Therefore, gaining a better understanding of bromine behaviour at high latitudes is important due to the impact it can potentially have on near surface ozone concentrations. The impact of such ozone





depletion on continental surface ozone across Western Siberia may be limited however, due to predominantly southerly winds in the north of the domain over the Siberian coast, during spring.

The dominant ozone dry deposition sink within the domain is to forest, with approximately half of this
5   deposition occurring north of 60°N to Arctic forest vegetation. This agrees well with the findings of who suggest that the Siberian forest is an important ozone surface sink through dry deposition. Our results show that summer (JJA) is the most active time for surface ozone deposition, correlating with peak photosynthetic activity and longer periods of stomatal opening, leading to more stomatal gas exchange. Stjernberg et al., (2012) also suggest that both tundra and wetlands are significant surface sinks for ozone.
10   Our findings support the importance of tundra, which is the second largest sink above 60°N behind forest land cover type. We find wetlands to have a small contribution to ozone deposition in our domain. However, we note that our domain is different and substantially smaller than the region considered by the Stjernberg et al., (2012) study.



## 5 Conclusions

We have used in-situ observations for surface ozone evaluation and OMI satellite observations of tropospheric column $NO_2$ for large spatial scale evaluation of ozone precursor distributions in the regional chemistry model WRF-Chem over Western Siberian during spring and summer. Gaining a better understanding of controls on tropospheric ozone concentrations in western Siberia is important due to the role it plays as a direct pathway to the Arctic. The region provides substantial surface sinks via efficient dry deposition to vegetation, and important sources for polluted air travelling polewards to the Arctic. We attempt to better quantify major sinks and sources of ozone and its precursors within this key region for high latitude and Arctic atmospheric composition, which is vastly understudied with limited in-situ observations. WRF-Chem shows an underestimation of tropospheric column $NO_2$ when compared with OMI, despite the use of a more recent OMI retrieval product (DOMINO v2), which has reduced a previously characterised high bias in OMI $NO_2$ columns in earlier product versions (Boersma et al., 2011). We suggest that the low model bias could be a result of lacking or underestimated emissions within the region in current emissions datasets, or due to model errors in the $NO_2$ atmospheric lifetime. Our results suggest that from May – August the simulated spatial pattern in $NO_2$ produced by the ECL anthropogenic emissions is consistent with observed $NO_2$ from OMI (R=0.61-0.74), but a persistent low bias continues throughout. Both EH2 (FMB= -0.82 to -0.73) and ECL (FMB= -0.80 to -0.70) produce simulated atmospheric distributions that underestimate the magnitude of satellite observed $NO_2$ during this period. Deficiencies in model tropospheric $NO_y$ chemistry have been identified in previous studies as a contributor to bias in the simulated $NO_x$ lifetime (Huijnen et al., 2010). These include removal through wet and dry deposition, and $NO_x$ too readily converted to reservoir species such as nitric acid and peroxyacetyl nitrate (PAN). Our larger model biases during summer could be an indication of errors in the conversion of $NO_2$ to nitric acid, when OH concentrations are enhanced and the $NO_2$+OH reaction is more important. Future work is needed to better understand drivers of the model $NO_2$ bias relative to OMI.

Our results suggest that surface ozone north of $60°N$ in the region studied is influenced by an interplay between seasonality in atmospheric transport patterns, vegetation dry deposition uptake and



photochemistry. We find that anthropogenic emissions have a more significant impact on surface ozone north of 60°N compared to fire emissions during our study period, with transport and energy emissions being the dominant ozone precursor sources.

5   Siberian forests act as an important surface sink to ozone, especially during June, July and August when ozone surface fluxes are largest, and account for 36% of all ozone deposition in this period (Fig. 10 & 11). With future northward migration of the treeline at high latitudes, understanding how this can act as a sink for ozone in the future is important, as this could go towards helping alleviate the high latitude tropospheric ozone burden.



*Code and data availability.* The code and data used in this study is available from the authors upon reasonable request.

*Supplement.*

*Author Contributions.* Conceptualization of research by TT and SRA, with support from DVS. TT
performed model simulations with support from LC. TT performed model evaluation and data analysis with support from SRA and RJP. CK provided WRFotron modelling scripts. TT wrote the manuscript. All authors provided comments on manuscript.

*Competing Interests.* The authors declare that they have no conflict of interests.

*Acknowledgements*. Thomas Thorp was funded by a studentship from the NERC SPHERES DTP (NE/L002574/1). This work was undertaken on ARC3, part of the High-Performance Computing facilities
at the University of Leeds, UK. We acknowledge the use of the WRFotron scripts developed by Christoph Knote to automatise WRF-Chem runs with re-initialised meteorology. Steve Arnold acknowledges support from the Belmont Forum and Natural Environment Research Council via the "Arctic Community Resilience to Boreal Environmental change: Assessing Risks from fire and disease (ACRoBEAR)" project (NE/T013672/1). Measurements at Fonovaya Observatory are supported by the Ministry of
Science and Higher Education of the Russian Federation (Programme No. AAAA-A17-117021310142-5). The work was supported by Academy of Finland via Centre of Excellence in Atmospheric Science (grant no. 307331) and NANOBIOMASS (307537) as well as 334792 (Belmont Forum project: Arctic Community Resilience to Boreal Environmental change: Assessing Risks from fire and disease (ACRoBEAR). In addition, the work was financially supported by European Commission through grant
agreement No 689443 via project iCUPE (Integrative and Comprehensive Understanding on Polar Environments) and by European Research Council (ATM-GTP).





AMAP: AMAP assessment 2015: Black carbon and ozone as Arctic climate forcers., 2015.

Antokhin, P. N., Arshinova, V. G., Arshinov, M. Y., Belan, B. D., Belan, S. B., Davydov, D. K., Ivlev, G. A., Fofonov, A. V., Kozlov, A. V., Paris, J. D., Nedelec, P., Rasskazchikova, T. M., Savkin, D. E., Simonenkov, D. V., Sklyadneva, T. K. and Tolmachev, G. N.: Distribution of Trace Gases and Aerosols in the Troposphere Over Siberia During Wildfires of Summer 2012, J. Geophys. Res. Atmos., 123(4), 2285–2297, doi:10.1002/2017JD026825, 2018.

Arnold, S., Law, K. S., Brock, C. A., Thomas, J. L., Starkweather, S. M., Salzen, K. vonB, Stohl, A., Sharma, S., Lund, M. T., Flanner, M. G., Petaja, T., Tanimoto, H., Gamble, J., Dibb, J. E., Melamed, M., Johnson, N., Fidel, M., Tynkkynen, V.-P., Baklanov, A., Eckhardt, S., Monks, S. a., Browse, J. and Bozem, H.: Arctic air pollution: Challenges and opportunities for the next decade, Elementa, 1–17, doi:10.12952/journal.elementa.000104, 2016.

Arnold, S. R., Lombardozzi, D., Lamarque, J. F., Richardson, T., Emmons, L. K., Tilmes, S., Sitch, S. A., Folberth, G., Hollaway, M. J. and Val Martin, M.: Simulated Global Climate Response to Tropospheric Ozone-Induced Changes in Plant Transpiration, Geophys. Res. Lett., 45(23), 13,070-13,079, doi:10.1029/2018GL079938, 2018.

Asmi, E., Kondratyev, V., Brus, D., Laurila, T., Lihavainen, H., Backman, J., Vakkari, V., Aurela, M., Hatakka, J., Viisanen, Y., Uttal, T., Ivakhov, V. and Makshtas, A.: Aerosol size distribution seasonal characteristics measured in Tiksi, Russian Arctic, Atmos. Chem. Phys., 16(3), 1271–1287, doi:10.5194/acp-16-1271-2016, 2016.

Atkinson, R. W., Butland, B. K., Dimitroulopoulou, C., Heal, M. R., Stedman, J. R., Carslaw, N., Jarvis, D., Heaviside, C., Vardoulakis, S., Walton, H. and Anderson, H. R.: Long-term exposure to ambient ozone and mortality: A quantitative systematic review and meta-analysis of evidence from cohort studies, BMJ Open, 6(2), 1–10, doi:10.1136/bmjopen-2015-009493, 2016.

Berchet, A., Paris, J. D., Ancellet, G., Law, K. S., Stohl, A., Nédélec, P., Arshinov, M. Y., Belan, B. D. and Ciais, P.: Tropospheric ozone over Siberia in spring 2010: Remote influences and stratospheric intrusion, Tellus, Ser. B Chem. Phys. Meteorol., 65(1), 0–14, doi:10.3402/tellusb.v65i0.19688, 2013.

Boersma, K. F., Eskes, H. J., Dirksen, R. J., Van Der A, R. J., Veefkind, J. P., Stammes, P., Huijnen, V., Kleipool, Q. L., Sneep, M., Claas, J., Leitão, J., Richter, A., Zhou, Y. and Brunner, D.: An improved



tropospheric NO2 column retrieval algorithm for the Ozone Monitoring Instrument, Atmos. Meas. Tech., 4(9), 1905–1928, doi:10.5194/amt-4-1905-2011, 2011.

Bucsela, E. J., Celarier, E. A., Wenig, M. O., Gleason, J. F., Veefkind, J. P., Boersma, K. F. and Brinksma, E. J.: Algorithm for NO2 vertical column retrieval from the ozone monitoring instrument, IEEE Trans. Geosci. Remote Sens., 44(5), 1245–1257, doi:10.1109/TGRS.2005.863715, 2006.

Cao, L., He, M., Jiang, H., Grosshans, H. and Cao, N.: Sensitivity of the reaction mechanism of the ozone depletion events during the arctic spring on the initial atmospheric composition of the troposphere, Atmosphere (Basel)., 7(10), doi:10.3390/atmos7100124, 2016.

Corbett, J. J., Lack, D. A., Winebrake, J. J. ., Harder, S. and Silberman, J. A.: Arctic shipping emissions inventories and future scenarios, Atmos. Chem. Phys., 10(19), 9689–9704, doi:10.5194/acp-10-9689-2010, 2010.

Crutzen, P. J., Lawrence, M. G. and Pöschl, U.: On the background photochemistry of tropospheric ozone, Tellus, Ser. A Dyn. Meteorol. Oceanogr., 51(1 SPEC. ISS.), 123–146, doi:10.1034/j.1600-0870.1999.t01-1-00010.x, 1999.

Davidson, E. A. and Kingerlee, W.: A global inventory of nitric oxide emissions from soils, Nutr. Cycl. Agroecosystems, 48(1–2), 37–50, doi:10.1023/A:1009738715891, 1997.

Ek, M. B.: Implementation of Noah land surface model advances in the National Centers for Environmental Prediction operational mesoscale Eta model, J. Geophys. Res., 108(D22), 1–16, doi:10.1029/2002jd003296, 2003.

Elvidge, C. D., Ziskin, D., Baugh, K. E., Tuttle, B. T., Ghosh, T., Pack, D. W., Erwin, E. H. and Zhizhin, M.: A fifteen year record of global natural gas flaring derived from satellite data, Energies, 2(3), 595–622, doi:10.3390/en20300595, 2009.

Emmons, L. K., Walters, S., Hess, P. G., Lamarque, J. F., Pfister, G. G., Fillmore, D., Granier, C., Guenther, A., Kinnison, D., Laepple, T., Orlando, J., Tie, X., Tyndall, G., Wiedinmyer, C., Baughcum, S. L. and Kloster, S.: Description and evaluation of the Model for Ozone and Related chemical Tracers, version 4 (MOZART-4), Geosci. Model Dev., 3(1), 43–67, doi:10.5194/gmd-3-43-2010, 2010.

Evans, M., Kholod, N., Malyshev, V., Tretyakova, S., Gusev, E., Yu, S. and Barinov, A.: Black carbon emissions from Russian diesel sources: Case study of Murmansk, Atmos. Chem. Phys., 15(14), 8349–





8359, doi:10.5194/acp-15-8349-2015, 2015.

Evans, M., Kholod, N., Kuklinski, T., Denysenko, A., Smith, S. J., Staniszewski, A., Hao, W. M., Liu, L. and Bond, T. C.: Black carbon emissions in Russia: A critical review, Atmos. Environ., 163, 9–21, doi:10.1016/j.atmosenv.2017.05.026, 2017.

Fuhrer, J.: Ozone risk for crops and pastures in present and future climates, Naturwissenschaften, 96(2), 173–194, doi:10.1007/s00114-008-0468-7, 2009.

Ganzeveld, L., Bouwman, L., Stehfest, E., Van Vuuren, D. P., Eickhout, B. and Lelieveld, J.: Impact of future land use and land cover changes on atmospheric chemistry-climate interactions, J. Geophys. Res. Atmos., 115(23), 1–18, doi:10.1029/2010JD014041, 2010.

Guenther, A., Karl, T., Harley, P., Wiedinmyer, C., Palmer, P. I. and C., G.: Estimates of global terrestrial isoprene emissions using MEGAN, Atmos. Chem. Phys. Discuss., 6(1), 107–173, doi:10.5194/acpd-6-107-2006, 2006.

Hains, J. C., Boersma, K. F., Mark, K., Dirksen, R. J., Cohen, R. C., Perring, A. E., Bucsela, E., Volten, H., Swart, D. P. J., Richter, A., Wittrock, F., Schoenhardt, A., Wagner, T., Ibrahim, O. W., Roozendael,

M. Van, Pinardi, G., Gleason, J. F., Veefkind, J. P. and Levelt, P.: Testing and improving OMI DOMINO tropospheric NO2 using observations from the DANDELIONS and INTEX-B validation campaigns, J. Geophys. Res. Atmos., 115(5), 1–20, doi:10.1029/2009JD012399, 2010.

Han, K., Lee, C. K., Lee, J., Kim, J. and Song, C. H.: A comparison study between model-predicted and OMI-retrieved tropospheric NO2 columns over the Korean peninsula, Atmos. Environ., 45(17), 2962–

2971, doi:10.1016/j.atmosenv.2010.10.016, 2011.

Herron-Thorpe, F. L., Lamb, B. K., Mount, G. H. and Vaughan, J. K.: Evaluation of a regional air quality forecast model for tropospheric NO2 columns using the OMI/Aura satellite tropospheric NO2 product, Atmos. Chem. Phys., 10(18), 8839–8854, doi:10.5194/acp-10-8839-2010, 2010.

Hodzic, A. and Jimenez, J. L.: Modeling anthropogenically controlled secondary organic aerosols in a

megacity: A simplified framework for global and climate models, Geosci. Model Dev., 4(4), 901–917, doi:10.5194/gmd-4-901-2011, 2011.

Hollaway, M. J., Arnold, S. R., Challinor, A. J. and Emberson, L. D.: Intercontinental trans-boundary contributions to ozone-induced crop yield losses in the Northern Hemisphere, Biogeosciences, 9(1), 271–





292, doi:10.5194/bg-9-271-2012, 2012.

Huang, K., Fu, J. S., Hodson, E. L., Dong, X., Cresko, J., Prikhodko, V. Y., Storey, J. M. and Cheng, M. D.: Identification of missing anthropogenic emission sources in Russia: Implication for modeling arctic haze, Aerosol Air Qual. Res., 14(7), 1799–1811, doi:10.4209/aaqr.2014.08.0165, 2014.

Huang, K., Fu, J. S., Prikhodko, V. Y., Storey, J. M., Romanov, A., Hodson, E. L., Cresko, J., Morozova, I., Ignatieva, Y. and Cabaniss, J.: Russian anthropogenic black carbon: Emission reconstruction and arctic black carbon simulation, J. Geophys. Res., 120(21), 11,306-11,333, doi:10.1002/2015JD023358, 2015.

Huijnen, V., Eskes, H. J., Poupkou, A., Elbern, H., Boersma, K. F., Foret, G., Sofiev, M., Valdebenito, A., Flemming, J., Stein, O., Gross, A., Robertson, L., D'Isidoro, M., Kioutsioukis, I., Friese, E., Amstrup,

B., Bergstrom, R., Strunk, A., Vira, J., Zyryanov, D., Maurizi, A., Melas, D., Peuch, V. H. and Zerefos, C.: Comparison of OMI NO2 tropospheric columns with an ensemble of global and European regional air quality models, Atmos. Chem. Phys., 10(7), 3273–3296, doi:10.5194/acp-10-3273-2010, 2010.

IPCC: Climate Change 2014 Synthesis Report, Contrib. Work. Groups I, II III to Fifth Assess. Rep. Intergov. Panel Clim. Chang., 1–112, doi:10.1017/CBO9781107415324, 2014.

Jaeglé, L., Steinberger, L., Martin, R. V. and Chance, K.: Global partitioning of NOx sources using satellite observations: Relative roles of fossil fuel combustion, biomass burning and soil emissions, Faraday Discuss., 130(x), 407–423, doi:10.1039/b502128f, 2005.

Jaffe, D. A. and Wigder, N. L.: Ozone production from wildfires: A critical review, Atmos. Environ., 51, 1–10, doi:10.1016/j.atmosenv.2011.11.063, 2012.

Janssens-Maenhout, G., Crippa, M., Guizzardi, D., Dentener, F., Muntean, M., Pouliot, G., Keating, T., Zhang, Q., Kurokawa, J., Wankmüller, R., Denier Van Der Gon, H., Kuenen, J. J. P., Klimont, Z., Frost, G., Darras, S., Koffi, B. and Li, M.: HTAP-v2.2: A mosaic of regional and global emission grid maps for 2008 and 2010 to study hemispheric transport of air pollution, Atmos. Chem. Phys., 15(19), 11411–11432, doi:10.5194/acp-15-11411-2015, 2015.

Jeong, J. I., Park, R. J. and Youn, D.: Effects of Siberian forest fires on air quality in East Asia during May 2003 and its climate implication, Atmos. Environ., 42(39), 8910–8922, doi:10.1016/j.atmosenv.2008.08.037, 2008.

Jerrett, M., Burnett, R. T., Arden Pope, C., Ito, K., Thurston, G., Krewski, D., Shi, Y., Calle, E. and Thun,





M.: Long-term ozone exposure and mortality, N. Engl. J. Med., 360(11), 1085–1095, doi:10.1056/NEJMoa0803894, 2009.

Jung, J., Lyu, Y., Lee, M., Hwang, T., Lee, S. and Oh, S.: Impact of Siberian forest fires on the atmosphere over the Korean Peninsula during summer 2014, Atmos. Chem. Phys., 16(11), 6757–6770,

doi:10.5194/acp-16-6757-2016, 2016.

Kanaya, Y., Irie, H., Takashima, H., Iwabuchi, H., Akimoto, H., Sudo, K., Gu, M., Chong, J., Kim, Y. J., Lee, H., Li, A., Si, F., Xu, J., Xie, P. H., Liu, W. Q., Dzhola, A., Postylyakov, O., Ivanov, V., Grechko, E., Terpugova, S. and Panchenko, M.: Long-term MAX-DOAS network observations of NO2 in Russia and Asia (MADRAS) during the period 2007-2012: Instrumentation, elucidation of climatology, and

comparisons with OMI satellite observations and global model simulations, Atmos. Chem. Phys., 14(15), 7909–7927, doi:10.5194/acp-14-7909-2014, 2014.

Karlsson, P. E., Tang, L., Sundberg, J., Chen, D., Lindskog, A. and Pleijel, H.: Increasing risk for negative ozone impacts on vegetation in northern Sweden, Environ. Pollut., 150(1), 96–106, doi:10.1016/j.envpol.2007.06.016, 2007.

Knote, C., Hodzic, A., Jimenez, J. L., Volkamer, R., Orlando, J. J., Baidar, S., Brioude, J., Fast, J., Gentner, D. R., Goldstein, A. H., Hayes, P. L., Knighton, W. B., Oetjen, H., Setyan, A., Stark, H., Thalman, R., Tyndall, G., Washenfelder, R., Waxman, E. and Zhang, Q.: Simulation of semi-explicit mechanisms of SOA formation from glyoxal in aerosol in a 3-D model, Atmos. Chem. Phys., 14(12), 6213–6239, doi:10.5194/acp-14-6213-2014, 2014.

Konovalov, I. B., Beekmann, M., Kuznetsova, I. N., Yurova, A. and Zvyagintsev, A. M.: Atmospheric impacts of the 2010 Russian wildfires: Integrating modelling and measurements of an extreme air pollution episode in the Moscow region, Atmos. Chem. Phys., 11(19), 10031–10056, doi:10.5194/acp-11-10031-2011, 2011.

Kukavskaya, E. A., Buryak, L. V., Shvetsov, E. G., Conard, S. G. and Kalenskaya, O. P.: The impact of

increasing fire frequency on forest transformations in southern Siberia, For. Ecol. Manage., 382, 225–235, doi:10.1016/j.foreco.2016.10.015, 2016.

Lamsal, L. N., Martin, R. V., Van Donkelaar, A., Celarier, E. A., Bucsela, E. J., Boersma, K. F., Dirksen, R., Luo, C. and Wang, Y.: Indirect validation of tropospheric nitrogen dioxide retrieved from the OMI





satellite instrument: Insight into the seasonal variation of nitrogen oxides at northern midlatitudes, J. Geophys. Res. Atmos., 115(5), 1–15, doi:10.1029/2009JD013351, 2010.

Law, K. S., Roiger, A., Thomas, J. L., Marelle, L., Raut, J. C., Dalsøren, S., Fuglestvedt, J., Tuccella, P., Weinzierl, B. and Schlager, H.: Local Arctic air pollution: Sources and impacts, Ambio, 46(s3), 453–463, doi:10.1007/s13280-017-0962-2, 2017.

Lelieveld, J., Evans, J. S., Fnais, M., Giannadaki, D. and Pozzer, A.: The contribution of outdoor air pollution sources to premature mortality on a global scale, Nature, 525(7569), 367–371, doi:10.1038/nature15371, 2015.

Li, C., Hsu, N. C., Sayer, A. M., Krotkov, N. A., Fu, J. S., Lamsal, L. N., Lee, J. and Tsay, S. C.: Satellite observation of pollutant emissions from gas flaring activities near the Arctic, Atmos. Environ., 133, 1–11, doi:10.1016/j.atmosenv.2016.03.019, 2016.

Marelle, L., Raut, J. C., Law, K. S. and Duclaux, O.: Current and Future Arctic Aerosols and Ozone From Remote Emissions and Emerging Local Sources—Modeled Source Contributions and Radiative Effects, J. Geophys. Res. Atmos., 1–22, doi:10.1029/2018JD028863, 2018.

Mölders, N., Tran, H. N. Q., Quinn, P., Sassen, K., Shaw, G. E. and Kramm, G.: Assessment of WRF/Chem to simulate sub-Arctic boundary layer characteristics during low solar irradiation using radiosonde, SODAR, and surface data, Atmos. Pollut. Res., 2(3), 283–299, doi:10.5094/APR.2011.035, 2011.

N. Kholod · M. Evans · T. Kuklinski: Russia's black carbon emissions: focus on diesel sources , Atmos. Chem. Phys., 2010(June), 1–27, doi:10.5194/acp-2016-475, 2016.

Oikawa, P. Y., Ge, C., Wang, J., Eberwein, J. R., Liang, L. L., Allsman, L. A., Grantz, D. A. and Jenerette, G. D.: Unusually high soil nitrogen oxide emissions influence air quality in a high-temperature agricultural region, Nat. Commun., 6, doi:10.1038/ncomms9753, 2015.

Pithan, F. and Mauritsen, T.: Arctic amplification dominated by temperature feedbacks in contemporary climate models, Nat. Geosci., 7(3), 181–184, doi:10.1038/ngeo2071, 2014.

Pope, R. J., Chipperfield, M. P., Savage, N. H., Ordóñez, C., Neal, L. S., Lee, L. A., Dhomse, S. S., Richards, N. A. D. and Keslake, T. D.: Evaluation of a regional air quality model using satellite column $NO_2$: treatment of observation errors and model boundary conditions and emissions, Atmos. Chem. Phys.,





15(10), 5611–5626, doi:10.5194/acp-15-5611-2015, 2015.

Quinn, P. K., Bates, T. S., Baum, E., Doubleday, N., Fiore, A. M., Flanner, M., Fridlind, A., Garrett, T. J., Koch, D., Menon, S., Shindell, D., Stohl, A. and Warren, S. G.: Short-lived pollutants in the Arctic: their climate impact and possible mitigation strategies, Atmos. Chem. Phys. Discuss., 7, 15669–15692, doi:10.5194/acpd-7-15669-2007, 2007.

Rydsaa, J. H., Stordal, F., Gerosa, G., Finco, A. and Hodnebrog: Evaluating stomatal ozone fluxes in WRF-Chem: Comparing ozone uptake in Mediterranean ecosystems, Atmos. Environ., 143, 237–248, doi:10.1016/j.atmosenv.2016.08.057, 2016.

Sand, M., Berntsen, T. K., von Salzen, K., Flanner, M. G., Langner, J. and Victor, D. G.: Response of Arctic temperature to changes in emissions of short-lived climate forcers, Nat. Clim. Chang., 6(November), 1–5, doi:10.1038/nclimate2880, 2015.

Sand, M., Berntsen, T. K., Von Salzen, K., Flanner, M. G., Langner, J. and Victor, D. G.: Response of Arctic temperature to changes in emissions of short-lived climate forcers, Nat. Clim. Chang., 6(3), 286–289, doi:10.1038/nclimate2880, 2016.

Schmale, J., Arnold, S. R., Law, K. S., Thorp, T., Anenberg, S., Simpson, W. R., Mao, J. and Pratt, K. A.: Local Arctic air pollution: A neglected but serious problem, Earth's Futur., 1–28, doi:10.1029/2018EF000952, 2018.

Shaposhnikov, D., Revich, B., Bellander, T., Bedada, G. B., Bottai, M., Kharkova, T., Kvasha, E., Lezina, E., Lind, T., Semutnikova, E. and Pershagen, G.: Mortality Related to Air Pollution with the Moscow Heat Wave and Wildfire of 2010, Epidemiology, 25(3), 359–364, doi:10.1097/EDE.0000000000000090, 2014.

Shindell, D., Kuylenstierna, J. C. I., Vignati, E., van Dingenen, R., Amann, M., Klimont, Z., Anenberg, S. C., Muller, N., Janssens-Maenhout, G., Raes, F., Schwartz, J., Faluvegi, G., Pozzoli, L., Kupiainen, K., Hoglund-Isaksson, L., Emberson, L., Streets, D., Ramanathan, V., Hicks, K., Oanh, N. T. K., Milly, G., Williams, M., Demkine, V. and Fowler, D.: Simultaneously Mitigating Near-Term Climate Change and Improving Human Health and Food Security, Science (80-. )., 335(6065), 183–189, doi:10.1126/science.1210026, 2012.

Sitch, S., Cox, P. M., Collins, W. J. and Huntingford, C.: Indirect radiative forcing of climate change





through ozone effects on the land-carbon sink., Nature, 448(August), 791–794, doi:10.1038/nature06059, 2007.

Sitnov, S. A., Mokhov, I. I. and Gorchakov, G. I.: The link between smoke blanketing of European Russia in summer 2016, Siberian wildfires and anomalies of large-scale atmospheric circulation, Dokl. Earth Sci., 472(2), 456–461, doi:10.1134/S1028334X17020052, 2017.

Skorokhod, A. I., Elansky, N. F., Belikov, I. B., Shumsky, R. A., Pankratova, N. V. and Lavrova, O. V.: Ozone and nitric oxides in the surface air over northern Eurasia according to observational data obtained in TROICA experiments, Izv. Atmos. Ocean. Phys., 47(3), 313–328, doi:10.1134/s0001433811030108, 2011.

Steinkamp, J. and Lawrence, M. G.: Improvement and evaluation of simulated global biogenic soil NO emissions in an AC-GCM, Atmos. Chem. Phys., 11(12), 6063–6082, doi:10.5194/acp-11-6063-2011, 2011.

Stevenson, D. S., Dentener, F. J., Schultz, M. G., Ellingsen, K., van Noije, T. P. C., Wild, O., Zeng, G., Amann, M., Atherton, C. S., Bell, N., Bergmann, D. J., Bey, I., Butler, T., Cofala, J., Collins, W. J., Derwent, R. G., Doherty, R. M., Drevet, J., Eskes, H. J., Fiore, A. M., Gauss, M., Hauglustaine, D. A., Horowitz, L. W., Isaksen, I. S. A., Krol, M. C., Lamarque, J. F., Lawrence, M. G., Montanaro, V., Müller, J. F., Pitari, G., Prather, M. J., Pyle, J. A., Rast, S., Rodriquez, J. M., Sanderson, M. G., Savage, N. H., Shindell, D. T., Strahan, S. E., Sudo, K. and Szopa, S.: Multimodel ensemble simulations of present-day and near-future tropospheric ozone, J. Geophys. Res. Atmos., 111(8), doi:10.1029/2005JD006338, 2006.

Stjernberg, A. C. E., Skorokhod, A., Paris, J. D., Elansky, N., Nédélec, P. and Stohl, A.: Low concentrations of near-surface ozone in Siberia, Tellus, Ser. B Chem. Phys. Meteorol., 64(1), doi:10.3402/tellusb.v64i0.11607, 2012.

Stohl, a., Berg, T., Burkhart, J. F., Fjæraa, a. M., Forster, C., Herber, a., Hov, Ø., Lunder, C., McMillan, W. W., Oltmans, S., Shiobara, M., Simpson, D., Solberg, S., Stebel, K., Ström, J., Tørseth, K., Treffeisen, R., Virkkunen, K. and Yttri, K. E.: Arctic smoke – record high air pollution levels in the European Arctic due to agricultural fires in Eastern Europe, Atmos. Chem. Phys. Discuss., 6(5), 9655–9722, doi:10.5194/acpd-6-9655-2006, 2006.

Stohl, A.: Characteristics of atmospheric transport into the Arctic troposphere, J. Geophys. Res. Atmos.,





111(11), 1–17, doi:10.1029/2005JD006888, 2006.

Stohl, A., Klimont, Z., Eckhardt, S., Kupiainen, K., Shevchenko, V. P., Kopeikin, V. M. and Novigatsky, A. N.: Black carbon in the Arctic: The underestimated role of gas flaring and residential combustion emissions, Atmos. Chem. Phys., 13(17), 8833–8855, doi:10.5194/acp-13-8833-2013, 2013.

Stohl, A., Aamaas, B., Amann, M., Baker, L. H., Bellouin, N., Berntsen, T. K., Boucher, O., Cherian, R., Collins, W., Daskalakis, N., Dusinska, M., Eckhardt, S., Fuglestvedt, J. S., Harju, M., Heyes, C., Hodnebrog, Hao, J., Im, U., Kanakidou, M., Klimont, Z., Kupiainen, K., Law, K. S., Lund, M. T., Maas, R., MacIntosh, C. R., Myhre, G., Myriokefalitakis, S., Olivié, D., Quaas, J., Quennehen, B., Raut, J. C., Rumbold, S. T., Samset, B. H., Schulz, M., Seland, Shine, K. P., Skeie, R. B., Wang, S., Yttri, K. E. and

Zhu, T.: Evaluating the climate and air quality impacts of short-lived pollutants, Atmos. Chem. Phys., 15(18), 10529–10566, doi:10.5194/acp-15-10529-2015, 2015.

Turner, M. C., Jerrett, M., Pope, C. A., Krewski, D., Gapstur, S. M., Diver, W. R., Beckerman, B. S., Marshall, J. D., Su, J., Crouse, D. L. and Burnett, R. T.: Long-Term Ozone Exposure and Mortality in a Large Prospective Study, Am. J. Respir. Crit. Care Med., 193(10), 1134–1142, doi:10.1164/rccm.201508-

1633OC, 2016.

Uttal, T., Starkweather, S., Drummond, J. R., Vihma, T., Makshtas, A. P., Darby, L. S., Burkhart, J. F., Cox, C. J., Schmeisser, L. N., Haiden, T., Maturilli, M., Shupe, M. D., De Boer, G., Saha, A., Grachev, A. A., Crepinsek, S. M., Bruhwiler, L., Goodison, B., McArthur, B., Walden, V. P., Dlugokencky, E. J., Persson, P. O. G., Lesins, G., Laurila, T., Ogren, J. A., Stone, R., Long, C. N., Sharma, S., Massling, A.,

Turner, D. D., Stanitski, D. M., Asmi, E., Aurela, M., Skov, H., Eleftheriadis, K., Virkkula, A., Platt, A., Førland, E. J., Iijima, Y., Nielsen, I. E., Bergin, M. H., Candlish, L., Zimov, N. S., Zimov, S. A., O'Neill, N. T., Fogal, P. F., Kivi, R., Konopleva-Akish, E. A., Verlinde, J., Kustov, V. Y., Vasel, B., Ivakhov, V. M., Viisanen, Y. and Intrieri, J. M.: International arctic systems for observing the atmosphere: An International Polar Year Legacy Consortium, Bull. Am. Meteorol. Soc., 97(6), 1033–1056,

doi:10.1175/BAMS-D-14-00145.1, 2016.

Vinken, G. C. M., Boersma, K. F., Maasakkers, J. D., Adon, M. and Martin, R. V.: Worldwide biogenic soil NOxemissions inferred from OMI NO2observations, Atmos. Chem. Phys., 14(18), 10363–10381, doi:10.5194/acp-14-10363-2014, 2014.





Visser, A. J., Boersma, K. F., Ganzeveld, L. N. and Krol, M. C.: European NO<sub><i>x</i></sub> emissions in WRF-Chem derived from OMI: impacts on summertime surface ozone, Atmos. Chem. Phys. Discuss., (2), 1–36, doi:10.5194/acp-2019-295, 2019.

Vivchar, A. V., Moiseenko, K. B., Shumskii, R. A. and Skorokhod, A. I.: Identifying anthropogenic

sources of nitrogen oxide emissions from calculations of Lagrangian trajectories and the observational data from a tall tower in Siberia during the spring-summer period of 2007, Izv. Atmos. Ocean. Phys., 45(3), 302–313, doi:10.1134/s0001433809030049, 2009.

Wesley, M. L.: Parameterization of Surface Resistances to Gaseous Dry Deposition in Regional-Scale Numerical Models, Atmos. Environ., 23(6), 1293–1304, doi:10.1016/S0950-351X(05)80241-1, 1989.

Wiedinmyer, C., Akagi, S. K., Yokelson, R. J., Emmons, L. K., Al-Saadi, J. A., Orlando, J. J. and Soja, A. J.: The Fire INventory from NCAR (FINN) – a high resolution global model to estimate the emissions from open burning, Geosci. Model Dev. Discuss., 3(4), 2439–2476, doi:10.5194/gmdd-3-2439-2010, 2010.

Yienger, J. J. and Levy, H.: Empirical model of global soil-biogenic NOxemissions, J. Geophys. Res.,

100(D6), doi:10.1029/95jd00370, 1995.

Zaveri, R. A., Easter, R. C., Fast, J. D. and Peters, L. K.: Model for Simulating Aerosol Interactions and Chemistry (MOSAIC), J. Geophys. Res. Atmos., 113(13), 1–29, doi:10.1029/2007JD008782, 2008.

Zhao, X., Strong, K., Adams, C., Schofield, R., Yang, X., Richter, A., Friess, U., Blechschmidt, A. M. and Koo, J. H.: A case study of a transported bromine explosion event in the Canadian high arctic, J.

Geophys. Res., 121(1), 457–477, doi:10.1002/2015JD023711, 2016.

Zhou, Y., Brunner, D., Boersma, K. F., Dirksen, R. and Wang, P.: An improved tropospheric NO2 retrieval for OMI observations in the vicinity of mountainous terrain, Atmos. Meas. Tech., 2(2), 401–416, doi:10.5194/amt-2-401-2009, 2009.

