# Peer review of "Late-Spring and Summertime Tropospheric Ozone and NO2 in Western Siberia and the Russian Arctic: Regional Model Evaluation and Sensitivities."

_Atmospheric Chemistry and Physics, 2020_

## Referee Comment (RC1) · Anonymous Referee #1 · 20 Aug 2020

General comments: The authors used regional air quality model WRF-Chem to investigate the processes controlling the regional distribution of tropospheric ozone over Western Siberia in late-spring and summer in 2011. They found that surface ozone in the region is controlled by an interplay between seasonality in atmospheric transport patterns, vegetation dry deposition, and a dominance of transport and energy sector emissions. Overall it is an interesting study. However, the presentations in particular of the figure qualities need to be substantially improved before it can be accepted to publish in ACP.

[Figure]

Major comments: 1. The quality of figures The figure colors or legends need to be carefully selected. Several figures are not clear. For example, very poor visual effect in Fig. 3 and Figure 7. Normally, the darker color may indicate the high concentrations, vice versa. However, the authors used an unusual color style. Page 14: hatching. I don't see any hatching. Not sure what the green colors mean. Fig. 7: the borders are too thick or too strong, making the shadings look less apparent. The color selections in Fig. 11 are very poor as well.

2. Page 15: There are strong biases of the WRF/Chem simulations. Any explanations? It is hard to believe the results with such strong biases.

Minor comments: The authors need to make a thorough check of the manuscript very carefully by eliminating the typos and mistakes.

1. Typos: Page14 to 15, "Figure 1", "Figure 2" should be changed to "Figure 3" and "Figure 4".

2. Page 13, line19 to 20, or example in Kazan, Perm, Yekaterinburg and Ufa It is better to have the locations mentioned in the manuscript marked in Fig. 3.

3. Page 16, line 5 to 9, "The transport sector is the dominant source for NOx in ECL and EH2 over Novosibirisk and Tomsk..." This information cannot be derived from Fig. 5 or other figures. Any source to support the statement?

4. Page 21, line 8, "North of 60°N the influence of high latitude gas flaring emissions is evident, which have greatest impact on NO2 in August (Fig. 7t)." This information cannot be reflected in Figure 7t.

5. Page 23, line 15, the latitude and longitude of "Ob valley" should be marked.

6. Page 23, line 17 to 18, "Surface ozone is most sensitive to anthropogenic emissions, particularly those from the transport sector (Fig 9)." cannot be clearly reflected in Figure 9.

7. Page 35, line 5, "Siberian" should be replaced by "Siberia".

8. For the units which are supposed to be superscript, the authors used subscripts sometimes. Please do the corrections.

---

## Referee Comment (RC2) · Anonymous Referee #2 · 3 Sep 2020

The manuscript presents a WRF-Chem modeling study of evaluating NOx emission inventories and ozone source attribution in West Siberia. The region has rapid changing emissions but few in situ measurements and thus the authors relied on OMI NO2 retrievals for inventory evaluation. The analysis is solid, and it is well written. I have a few comments on the modeling approach.

1) When is the OMI overpass time over West Siberia? Did the authors sample the WRF-Chem outputs at the time of OMI overpass and remove model days when no data from OMI is available (e.g. due to clouds), or simply used the model's monthly

mean for comparison? The former should be the correct way. There is no mention of this in the manuscript.

2) For the sensitivity simulations of zeroing transportation, energy, and fire emissions, did the authors turn off only NOx emissions or were other emissions (e.g. VOCs) from these sectors also turned off? It is not clear in the manuscript. Since the sensitivity is per sector, all emissions from the sector should be turned off.

3) The model using either of the two inventories underestimates NO2 columns in cities by a factor of two in the warm season (May – August), as shown by Figure 2. This large bias suggests there is a large missing source of NOx in the region or a large underestimate in some sectors' emissions. Without correction for the low bias, the model's sensitivity analysis of sector's contribution should not be reliable. The authors should estimate the impact of the low bias on their source attributions. One way to do that is to run another sensitivity analysis of increasing NOx emissions in the model to match with OMI NO2 columns and compare the resulting changes in NOx and ozone to the sector's contributions.

Minor comments: 1) pg 14-15: these figures should be Figure 3 and Figure 4.

2) pg 8, section 2.2: What does "anthropogenic" to soil NOx emissions refer to? Did the model include "non-anthropogenic" component of soil NOx emissions?

---

## Author Comment (AC1) · 18 Dec 2020

**"Late-spring and summertime tropospheric ozone and NO$_2$ in Western Siberia and the Russian Arctic: Regional model evaluation and sensitivities".**

**Response to Reviewer Comments**

We thank the two anonymous reviewers for their assessment of our manuscript and their constructive comments. Having addressed these, we feel they have led to improvements in our paper. Below we address each of the Reviewer comments in turn. Reviewer text is shown in italics, and our responses are shown in normal typeface. Where additional text added to the manuscript is shown, this is displayed in bold.

**Response to Reviewer 1**

*Major comments*

*1. The quality of figures The figure colors or legends need to be carefully selected. Several figures are not clear. For example, very poor visual effect in Fig. 3 and Figure 7. Normally, the darker color may indicate the high concentrations, vice versa. However, the authors used an unusual color style. Page 14: hatching. I don't see any hatching. Not sure what the green colors mean. Fig. 7: the borders are too thick or too strong, making the shadings look less apparent. The color selections in Fig. 11 are very poor as well.*

We agree with the reviewer that some of our figures were not as clear as they could be. We have modified several figures to improve clarity.

Panel sizes in multi-panel contour map plots (Figs. 3, 7, 8, 10) have been increased by removing excess whitespace, enabled by removing repetition of longitude and latitude labels and moving panel labels into the panels.

Hatching in Figure 3 has been removed as we agree that it was not possible to see this clearly in the small panel sizes. Instead, we have added a plot to the supplementary material (Fig. S1), to show locations where model-OMI differences exceed OMI uncertainties.

Despite the reviewer comment regarding the colour scale used, we have retained this. The colour scale in question (viridis) is linear in its colour spacing and avoids issues with red-green colour blindness. This colour scale has been highlighted as a good example to use for contour plots in a recent article on science communication (Crameri et al., 2020). We feel that removal of the hatching and increase in panel size has resulted in improved clarity of the figures.

Finally, we have replotted Figure 4 scatter plots using logarithmic axes scales. This adds clarity to the group of points in the lower NO$_2$ column range, allowing a better evaluation of the model performance over the full range of column values. We have added 1:2 and 2:1 lines to these plots to allow indicative assessment of the overall simulation bias.

*2. Page 15: There are strong biases of the WRF/Chem simulations. Any explanations? It is hard to believe the results with such strong biases.*

We agree that the large bias in the WRF-Chem simulated NO$_2$ warrants further examination. Reviewer 2 also made a similar point. We refer Reviewer 1 to our response to Reviewer 2 on this issue for information regarding how we have addressed this.

*Minor comments: The authors need to make a thorough check of the manuscript very carefully by eliminating the typos and mistakes.*

We apologise that several typographical errors were present in the submitted manuscript. We have thoroughly checked over the resubmitted text to eliminate these.

*1. Typos: Page14 to 15, "Figure 1", "Figure 2" should be changed to "Figure 3" and "Figure 4".*

Corrected.

*2. Page 13, line19 to 20, or example in Kazan, Perm, Yekaterinburg and Ufa It is better to have the locations mentioned in the manuscript marked in Fig. 3.*

We have added longitude and latitude labels to these locations mentioned in the text.

*3. Page 16, line 5 to 9, "The transport sector is the dominant source for NOx in ECL and EH2 over Novosibirsk and Tomsk. . ." This information cannot be derived from Fig. 5 or other figures. Any source to support the statement?*

Apologies. We quote this information from our analysis of the emissions datasets themselves. We have clarified this in the text:

"**Examination of sector totals in the ECL and EH2 emissions datasets shows that** the transport sector is the dominant source for $NO_x$ in ECL and EH2 over Novosibirisk and Tomsk..."

*4. Page 21, line 8, "North of 60°N the influence of high latitude gas flaring emissions is evident, which have greatest impact on NO2 in August (Fig. 7t)." This information cannot be reflected in Figure 7t.*

We have reworded this for clarity:

**"Hotspots in OMI-observed $NO_2$ north of 60°N in the central and western portions of the domain are associated with the influence of high latitude gas flaring emissions and are evident as substantial reductions in the ene_off simulation**

*5. Page 23, line 15, the latitude and longitude of "Ob valley" should be marked.*

This has been added in on first mention of the Ob Valley region (Page 3, line 15).

*6. Page 23, line 17 to 18, "Surface ozone is most sensitive to anthropogenic emissions, particularly those from the transport sector (Fig 9)." cannot be clearly reflected in Figure 9.*

We agree that this is not clear. We have reworded lines 17-19 to read:

**"Overall, surface ozone shows greatest sensitivity to anthropogenic emissions (Fig. 9). In June, July and August, transport sector emissions produce the largest ozone sensitivity, while energy emissions dominate during May.**

*7. Page 35, line 5, "Siberian" should be replaced by "Siberia".*

Corrected.

*8. For the units which are supposed to be superscript, the authors used subscripts sometimes. Please do the corrections.*

Corrected.

**Response to Reviewer 2**

*1) When is the OMI overpass time over West Siberia? Did the authors sample the WRF-Chem outputs at the time of OMI overpass and remove model days when no data from OMI is available (e.g. due to clouds), or simply used the model's monthly mean for comparison? The former should be the correct way. There is no mention of this in the manuscript.*

Apologies that this important information was missing. We have added the following information to Section 2.3.2:

**"Retrievals with geometric cloud cover greater than 20% and poor-quality data flags (where flag =-1) were removed. We compare model output and satellite observations on days where OMI data was available at the satellite overpass time (1330 local time)."** (Page 11, line 4-6)

*2) For the sensitivity simulations of zeroing transportation, energy, and fire emissions, did the authors turn off only NOx emissions or were other emissions (e.g. VOCs) from these sectors also turned off? It is not clear in the manuscript. Since the sensitivity is per sector, all emissions from the sector should be turned off.*

All emissions for a given sector were turned off. We have clarified this point by modifying lines 6-9 on Page 12:

"Three separate sensitivity simulations are conducted, within each of which **emissions of all species from** one of three different emission **sectors** are removed: biomass burning emissions (fires_off simulation), anthropogenic transport emissions (trans_off simulation), and anthropogenic energy emissions (ene_off simulation)."

*3) The model using either of the two inventories underestimates NO2 columns in cities by a factor of two in the warm season (May – August), as shown by Figure 2. This large bias suggests there is a large missing source of NOx in the region or a large underestimate in some sectors' emissions. Without correction for the low bias, the model's sensitivity analysis of sector's contribution should not be reliable. The authors should estimate the impact of the low bias on their source attributions. One way to do that is to run another sensitivity analysis of increasing NOx emissions in the model to match with OMI NO2 columns and compare the resulting changes in NOx and ozone to the sector's contributions.*

We thank the reviewer for highlighting this issue, and for suggesting a potential way forward to deal with the large $NO_2$ bias in the model simulations. The large bias was also highlighted by Reviewer 1. In light of this comment, we have undertaken additional model simulations to investigate the bias.

Firstly, having examined the data coverage in the context of OMI uncertainty and model bias we have decided to remove April from the analysis, since the coverage is poorer than in other months at high latitudes. This does not affect the overall conclusions and context for our results, and simplifies the discussion of model bias to months where there is consistent

data coverage. Throughout the paper we have therefore removed April from the plots and where it appears in discussion of the results.

As suggested by the reviewer, we have investigated the sensitivity of the model $NO_2$ bias to emissions. Since the model simulations are computationally intensive, it has not been possible to run with multiple different scaling assumptions. We elected to scale the dominant anthropogenic sector emissions (transport and energy) in the ECLIPSE inventory together by a factor of 2 (accounting for 82% of anthropogenic NOx emissions), to test the sensitivity over the 4-month simulation. These scaled emissions substantially reduce the mean $NO_2$ bias in several of the urban regions of the domain (see updated Figure 5), although increase the positive bias in some of the regions. Overall the model $NO_2$ performance is improved, particularly over anthropogenic source regions (see scatter plot Fig. 4, and new Table 2). However, we find the low $NO_2$ bias in background regions remains similar, during summer in particular. We note that model bias in much of this background region in the domain is lower than the OMI observational error (see new Fig S1), meaning that it is difficult to draw conclusions regarding the bias in these regions. We have added a new Table 2 to the main manuscript (Section 3.1) to show mean monthly RMSE error for simulations with the two control emission datasets and the scaled emissions over urban and background regions to highlight how the model error responds to the different emissions.

Given the improvement in the model simulation on average with the scaled emissions, we take these forward as the control simulation and the basis of the sensitivity simulations for the ozone attribution analysis in Section 3.3. For this, we have produced new emissions sensitivity simulations based on the improved control emissions. We note that the ozone model evaluation against observations is marginally worse with the scaled emissions (Fig. 6), however given the limited spatial coverage of the ozone data we base the optimal simulation choice on the more extensive $NO_2$ evaluation. In the text we now present ozone source attribution values based on these new scaled emissions. We also present plots for the experiments using the standard emissions in supplementary for comparison (new Figs. S2, S3, S5, S6).

*Minor comments:*

*1) pg 14-15: these figures should be Figure 3 and Figure 4.*

Apologies for the mis-labelling. This has been corrected.

*2) pg 8, section 2.2: What does "anthropogenic" to soil NOx emissions refer to? Did the model include "non-anthropogenic" component of soil NOx emissions?*

To clarify – we have added anthropogenic NOx emissions related to fertilized agricultural soils, which are absent from the ECLIPSE emission dataset and not included with natural soil NOx emissions. We have clarified this by modifying text on Page 8, Line 22:

"**Past studies have highlighted potential missing sources of anthropogenic soil NOx emissions in current inventories, associated with fertilized agricultural soils.**"

**References**

Crameri, F., Shephard, G.E. & Heron, P.J. The misuse of colour in science communication.*Nat Commun* **11,** 5444 (2020). https://doi.org/10.1038/s41467-020-19160-7